# SELF-SUPERVISED INFERENCE IN STATE-SPACE MODELS

**David Ruhe**
AI4Science, AMLab, Anton Pannekoek Institute
University of Amsterdam, The Netherlands
d.ruhe@uva.nl

**Patrick Forré**
AI4Science, AMLab
University of Amsterdam, The Netherlands
p.d.forre@uva.nl

## ABSTRACT

We perform approximate inference in state-space models with nonlinear state transitions. Without parameterizing a generative model, we apply Bayesian update formulas using a local linearity approximation parameterized by neural networks. This comes accompanied by a maximum likelihood objective that requires no supervision via uncorrupt observations or ground truth latent states. The optimization backpropagates through a recursion similar to the classical Kalman filter and smoother. Additionally, using an approximate conditional independence, we can perform smoothing without having to parameterize a separate model. In scientific applications, domain knowledge can give a linear approximation of the latent transition maps, which we can easily incorporate into our model. Usage of such domain knowledge is reflected in excellent results (despite our model's simplicity) on the chaotic Lorenz system compared to fully supervised and variational inference methods. Finally, we show competitive results on an audio denoising experiment.

## 1 INTRODUCTION

Many sequential processes in industry and research involve noisy measurements that describe latent dynamics. A state-space model is a type of graphical model that effectively represents such noise-afflicted data (Bishop, 2006). The joint distribution is assumed to factorize according to a directed graph that encodes the dependency between variables using conditional probabilities. One is usually interested in performing inference, meaning to obtain reasonable estimates of the posterior distribution of the latent states or uncorrupt measurements. Approaches involving sampling (Neal et al., 2011), variational inference (Kingma & Welling, 2013), or belief propagation (Koller & Friedman, 2009) have been proposed before. Assuming a hidden Markov process (Koller & Friedman, 2009), the celebrated Kalman filter and smoother (Kalman, 1960; Rauch et al., 1965) are classical approaches to solving the posterior inference problem. However, the Markov assumption, together with linear Gaussian transition and emission probabilities, limit their flexibility. We present filtering and smoothing methods that are related to the classical Kalman filter updates but are augmented with flexible function estimators without using a constrained graphical model. By noting that the filtering and smoothing recursions can be back-propagated through, these estimators can be trained with a principled maximum-likelihood objective reminiscent of the `noise2noise` objective (Lehtinen et al., 2018; Laine et al., 2019). By using a locally linear transition distribution, the posterior distribution remains tractable despite the use of non-linear function estimators. Further, we show how a *linearized smoothing* procedure can be applied directly to the filtering distributions, discarding the need to train a separate model for smoothing.

To verify what is claimed, we perform three experiments. (1) A linear dynamics filtering experiment, where we show how our models approximate the optimal solution with sufficient data. We also report that including expert knowledge can yield better estimates of latent states. (2) A more challenging chaotic Lorenz smoothing experiment that shows how our models perform on par with recently proposed supervised models. (3) An audio denoising experiment that uses real-world noise showing practical applicability of the methods.

Our contributions can be summarized as follows.

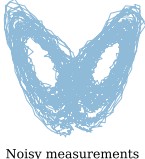 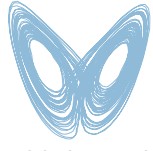 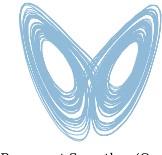 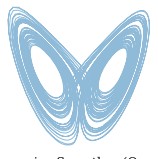 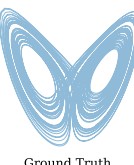

Noisy measurements     Extended Kalman Smoother     Recurrent Smoother (Ours)     Recursive Smoother (Ours)     Ground Truth

Figure 1: Best viewed on screen. Qualitative results of our work. To the noisy measurements (1st from left), we apply an extended Kalman smoother (2nd). From the noisy measurements, we learn a recurrent model that does slightly better (3rd). Our recursive model combines expert knowledge with inference (4th), yielding the best result. Ground truth provided for comparison (5th).

1. We show that the posterior inference distribution of a state-space model is tractable while parameter estimation is performed by neural networks. This means that we can apply the classical recursive Bayesian updates, akin to the Kalman filter and smoother, with mild assumptions on the generative process.

2. Our proposed method is optimized using maximum likelihood in a self-supervised manner. That is, ground truth values of states and measurements are not assumed to be available for training. Still, despite our model's simplicity, our experiments show that it performs better or on par with several baselines.

3. We show that the model can be combined with prior knowledge about the transition and emission probabilities, allowing for better applicability in low data regimes and incentivizing the model to provide more interpretable estimates of the latent states.

4. A linearized smoothing approach is presented that does not require explicit additional parameterization and learning of the smoothing distribution.

## 2 RELATED WORK

Becker et al. (2019) provide a detailed discussion of recent related work, which we build on here and in table 1. An early method that extends the earlier introduced Kalman filter by allowing nonlinear transitions and emissions is the Extended Kalman filter (Ljung, 1979). It is limited due to the naive approach to locally linearize the transition and emission distributions. Furthermore, the transition and emission mechanisms are usually assumed to be known, or estimated with Expectation Maximization (Moon, 1996). More flexible methods that combine deep learning with variational inference include Black Box Variational Inference (Archer et al., 2015), Structured Inference Networks (Krishnan et al., 2017), Kalman Variational Autoencoder (Fraccaro et al., 2017), Deep Variational Bayes Filters (Karl et al., 2017), Variational Sequential Monte Carlo (Naesseth et al., 2018) and Disentangled Sequential Autoencoder (Yingzhen & Mandt, 2018). However, the lower-bound objective makes the approach less scalable and accurate (see also Becker et al. (2019). Furthermore, all of the above methods explicitly assume a graphical model, imposing a strong but potentially harmful inductive bias. The BackpropKF (Haarnoja et al., 2016) and Recurrent Kalman Network (Becker et al., 2019) move away from variational inference and borrow Bayesian filtering techniques from the Kalman filter. We follow this direction but do not require supervision through ground truth latent states or uncorrupt emissions. Satorras et al. (2019) combine Kalman filters through message passing with graph neural networks to perform hybrid inference. We perform some of their experiments by also incorporating expert knowledge. However, contrary to their approach, we do not need supervision. Finally, concurrently to this work, Revach et al. (2021) develop KalmanNet. It proposes similar techniques but evaluates them in a supervised manner. The authors, however, do suggest that an unsupervised approach can also be feasible. Additionally, we more explicitly state what generative assumptions are required, then target the posterior distribution of interest, and develop the model and objective function from there. Moreover, the current paper includes linearized smoothing (section 6), parameterized smoothing (appendix A), and the recurrent model (appendix C). We also denote theoretical guarantees under the `noise2noise` objective.

|  | scalable | state est. | uncertainty | noise | dir. opt. | self-sup. |
|---|---|---|---|---|---|---|
| Ljung (1979) | ✓ / × | ✓ | ✓ | ✓ | × | × |
| Hochreiter et. al. (1997) | ✓ | ✓ | ✓ / × | ✓ | ✓ | × |
| Cho et al. (2014) | ✓ | ✓ | ✓ / × | ✓ | ✓ | × |
| Wahlström et al. (2015) | ✓ | ✓ | ✓ / × | × | ✓ | × |
| Watter et al. (2015) | ✓ | × | ✓ | ✓ | × | ✓ |
| Archer et al. (2015) | ✓ / × | × | ✓ | ✓ | × | ✓ |
| Krishnan et al. (2017) | ✓ | × | ✓ | ✓ | × | ✓ |
| Fraccaro et al. (2017) | ✓ / × | × | ✓ | ✓ | × | ✓ |
| Karl et al. (2017) | ✓ | × | ✓ | ✓ | × | ✓ |
| Naesseth et al. (2018) | ✓ | × | ✓ | ✓ | × | ✓ |
| Yingzhen et al. (2018) | ✓ | × | ✓ | × | × | ✓ |
| Rangapuram et al. (2018) | ✓ / × | ✓ (1D) | ✓ | × | ✓ | ✓ |
| Doerr et al. (2018) | × | ✓ | ✓ | ✓ | ✓ | ✓ |
| Satorras et al. (2019) | ✓ | ✓ | × | ✓ | ✓ | × |
| Haarnoja et al. (2016) | ✓ | ✓ | ✓ | ✓ | ✓ | × |
| Becker et al. (2019) | ✓ | ✓ | ✓ | ✓ | ✓ | × |
| **Ours** | ✓ / × | ✓ | ✓ | ✓ | ✓ | ✓ |

Table 1: We compare whether algorithms are scalable, state estimation can be performed, models provide uncertainty estimates, noisy or missing data can be handled, optimization is performed directly and if supervision is required. "✓ / ×" means that it depends on the parameterization.

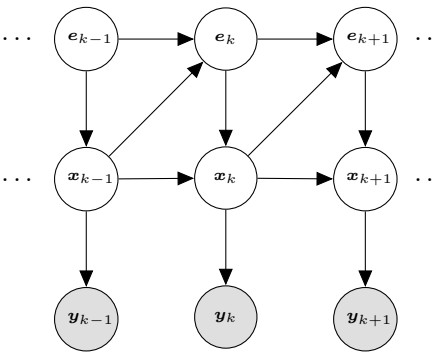

Figure 2: State-space model with deeper latent structure.

## 3 GENERATIVE MODEL ASSUMPTIONS

In this section, we explicitly state the model's generative process assumptions. First, we assume that we can measure (at least) one run of (noise-afflicted) sequential data $\boldsymbol{y}_{0:K} := (\boldsymbol{y}_0, \ldots, \boldsymbol{y}_K)$, where each $\boldsymbol{y}_k \in \mathbb{R}^M$, $k = 0, \ldots, K$. We abbreviate: $\boldsymbol{y}_{l:k} := (\boldsymbol{y}_l, \ldots, \boldsymbol{y}_k)$ and $\boldsymbol{y}_{<k} := \boldsymbol{y}_{0:k-1}$ and $\boldsymbol{y}_{\leq k} := \boldsymbol{y}_{0:k}$ and $\boldsymbol{y}_{-k} := (\boldsymbol{y}_{0:k-1}, \boldsymbol{y}_{k+1:K})$. We then assume that $\boldsymbol{y}_{0:K}$ is the result of some possibly non-linear probabilistic latent dynamics, i.e., of a distribution $p(\boldsymbol{x}_{0:K})$, whose variables are given by $\boldsymbol{x}_{0:K} := (\boldsymbol{x}_0, \ldots, \boldsymbol{x}_K)$ with $\boldsymbol{x}_k \in \mathbb{R}^N$. Each $\boldsymbol{y}_k$ is assumed to be drawn from some shared noisy emission probability $p(\boldsymbol{y}_k \mid \boldsymbol{x}_k)$. The joint probability is then assumed to factorize as:

$$p(\boldsymbol{y}_{0:K}, \boldsymbol{x}_{0:K}) = p(\boldsymbol{x}_{0:K}) \prod_{k=0}^{K} p(\boldsymbol{y}_k \mid \boldsymbol{x}_k). \tag{1}$$

Further implicit assumptions about the generative model are imposed via inference model choices (see section 7). Note that this factorization encodes several conditional independences like

$$\boldsymbol{y}_k \perp\!\!\!\perp (\boldsymbol{y}_{-k}, \boldsymbol{x}_{-k}) \mid \boldsymbol{x}_k. \tag{2}$$

Typical models that follow these assumptions are linear dynamical systems, hidden Markov models, but also nonlinear state-space models with higher-order Markov chains in latent space like presented in fig. 2.

In contrast to other approaches (e.g., Krishnan et al. (2015); Johnson et al. (2016); Krishnan et al. (2017)) where one tries to model the latent dynamics with transition probabilities $p(\boldsymbol{x}_k \mid \boldsymbol{x}_{k-1})$ and possibly non-linear emission probabilities $p(\boldsymbol{y}_k \mid \boldsymbol{x}_k)$, we go the other way around. We assume that all the non-linear dynamics are captured inside the latent distribution $p(\boldsymbol{x}_{0:K})$, where at this point we make no further assumption about its factorization, and the emission probabilities are (well-approximated with) a linear Gaussian noise model:

$$p(\boldsymbol{y}_k \mid \boldsymbol{x}_k) = \mathcal{N}(\boldsymbol{y}_k \mid \boldsymbol{H}\boldsymbol{x}_k, \boldsymbol{R}), \tag{3}$$

where the matrix $\boldsymbol{H}$ represents the measurement device and $\boldsymbol{R}$ is the covariance matrix of the independent additive noise. We make a brief argument why this assumption is not too restrictive. First, if one is interested in denoising corrupted measurements, any nonlinear emission can be captured directly inside the latent states $\boldsymbol{x}_k$. To see this, let $\boldsymbol{z}_k \in \mathbb{R}^{N-M}$ denote non-emitted state variables. We then put $\boldsymbol{x}_k := [\boldsymbol{y}_k \quad \boldsymbol{z}_k]^\top$, where $\boldsymbol{y}_k$ is computed by applying the nonlinear emission to $\boldsymbol{z}_k$. We thus include the measurements in the modeled latent state $\boldsymbol{x}_k$. Then we can put $\boldsymbol{H} := [\boldsymbol{I}_M \quad \boldsymbol{0}_{M \times (N-M)}]$. Second, techniques proposed by Laine et al. (2019) allow for non-Gaussian noise models, relaxing the need for assumption eq. (3). Third, we can locally linearize the

emission (Ljung, 1979). Finally, industrial or academic applications include cases where emissions are (sparse) Gaussian measurements and the challenging nonlinear dynamics occur in latent space. Examples can be found in MRI imaging (Lustig et al., 2007) and radio astronomy (Thompson et al., 2017).

## 4 PARAMETERIZATION

In this section, we show how we parameterize the inference model. A lot of the paper's work relies on established Bayesian filtering machinery. However, for completeness, we like to prove how all the update steps remain valid while using neural networks for function estimation.

Given our noisy measurements $\boldsymbol{y}_{0:K} = (\boldsymbol{y}_0, \ldots, \boldsymbol{y}_K)$ we want to find good estimates for the latent states $\boldsymbol{x}_{0:K} = (\boldsymbol{x}_0, \ldots, \boldsymbol{x}_K)$, which generated $\boldsymbol{y}_{0:K}$. For this, we want to infer the marginal conditional distributions $p(\boldsymbol{x}_k \mid \boldsymbol{y}_{\leq k})$ or $p(\boldsymbol{x}_k \mid \boldsymbol{y}_{<k})$ (for forecasting), for an online inference approach (*filtering*); and $p(\boldsymbol{x}_k \mid \boldsymbol{y}_{0:K})$ or $p(\boldsymbol{x}_k \mid \boldsymbol{y}_{-k})$, for a full inference approach (*smoothing*). In the main body of the paper, we only consider filtering. Smoothing can be performed similarly, which is detailed in the supplementary material (appendix A).

We start with the following advantageous parameterization:

$$p(\boldsymbol{x}_k \mid \boldsymbol{x}_{k-1}, \boldsymbol{y}_{<k}) = \mathcal{N}\left(\boldsymbol{x}_k \mid \hat{\boldsymbol{F}}_{k|<k}\,\boldsymbol{x}_{k-1} + \hat{e}_{k|<k}, \hat{\boldsymbol{Q}}_{k|<k}\right), \tag{4}$$

where $\hat{\boldsymbol{F}}_{k|<k} := \hat{\boldsymbol{F}}_{k|<k}(\boldsymbol{y}_{<k})$, $\hat{e}_{k|<k} := \hat{e}_{k|<k}(\boldsymbol{y}_{<k})$ and $\hat{\boldsymbol{Q}}_{k|<k} := \hat{\boldsymbol{Q}}_{k|<k}(\boldsymbol{y}_{<k})$ are parameterized with neural networks. Next, we have available

$$p(\boldsymbol{x}_{k-1} \mid \boldsymbol{y}_{\leq(k-1)}) = \mathcal{N}\left(\boldsymbol{x}_{k-1} \mid \hat{\boldsymbol{x}}_{(k-1)|\leq(k-1)}, \hat{\boldsymbol{P}}_{(k-1)|\leq(k-1)}\right), \tag{5}$$

i.e., the previous time-step's conditional marginal distribution of interest. For $k = 1$, this is some initialization. Otherwise, it is the result of the procedure we are currently describing. We use this distribution to evaluate the marginalization

$$\begin{aligned} p(\boldsymbol{x}_k \mid \boldsymbol{y}_{<k}) &= \int p(\boldsymbol{x}_k \mid \boldsymbol{x}_{k-1}, \boldsymbol{y}_{<k})\, p(\boldsymbol{x}_{k-1} \mid \boldsymbol{y}_{<k})\, d\boldsymbol{x}_{k-1} \\ &= \mathcal{N}\left(\boldsymbol{x}_k \mid \hat{\boldsymbol{x}}_{k|<k}, \hat{\boldsymbol{P}}_{k|<k}\right), \end{aligned} \tag{6}$$

with

$$\hat{\boldsymbol{x}}_{k|<k}(\boldsymbol{y}_{<k}) = \hat{\boldsymbol{F}}_{k|<k}\,\hat{\boldsymbol{x}}_{k-1|\leq k-1} + \hat{e}_{k|<k}, \quad \hat{\boldsymbol{P}}_{k|<k}(\boldsymbol{y}_{<k}) = \hat{\boldsymbol{F}}_{k|<k}\,\hat{\boldsymbol{P}}_{k-1|\leq k-1}\,\hat{\boldsymbol{F}}_{k|<k}^{\top} + \hat{\boldsymbol{Q}}_{k|<k}. \tag{7}$$

Note that the distributions under the integral eq. (6) are jointly Gaussian only because of the parameterization eq. (4). Hence, we can evaluate the integral analytically.

Finally, to obtain the conditional $p(\boldsymbol{x}_k \mid \boldsymbol{y}_{\leq k}) = p(\boldsymbol{x}_k \mid \boldsymbol{y}_k, \boldsymbol{y}_{<k})$ we use Bayes' rule:

$$p(\boldsymbol{x}_k \mid \boldsymbol{y}_k, \boldsymbol{y}_{<k}) = \frac{p(\boldsymbol{y}_k \mid \boldsymbol{x}_k, \boldsymbol{y}_{<k}) \cdot p(\boldsymbol{x}_k \mid \boldsymbol{y}_{<k})}{p(\boldsymbol{y}_k \mid \boldsymbol{y}_{<k})} \overset{\text{eq. (2)}}{=} \frac{p(\boldsymbol{y}_k \mid \boldsymbol{x}_k)}{p(\boldsymbol{y}_k \mid \boldsymbol{y}_{<k})} \cdot p(\boldsymbol{x}_k \mid \boldsymbol{y}_{<k}). \tag{8}$$

Equation (3) and the result eq. (6) allow us to also get an analytic expression for

$$p(\boldsymbol{x}_k \mid \boldsymbol{y}_{\leq k}) = \mathcal{N}\left(\boldsymbol{x}_k \mid \hat{\boldsymbol{x}}_{k|\leq k}, \hat{\boldsymbol{P}}_{k|\leq k}\right) \tag{9}$$

with the following abbreviations:

$$\hat{\boldsymbol{x}}_{k|\leq k}(\boldsymbol{y}_{\leq k}) := \hat{\boldsymbol{x}}_{k|<k} + \hat{\boldsymbol{K}}_k\,(\boldsymbol{y}_k - \boldsymbol{H}\,\hat{\boldsymbol{x}}_{k|<k}), \quad \hat{\boldsymbol{P}}_{k|\leq k}(\boldsymbol{y}_{\leq k}) := \hat{\boldsymbol{P}}_{k|<k} - \hat{\boldsymbol{K}}_k\,\boldsymbol{H}\,\hat{\boldsymbol{P}}_{k|<k}. \tag{10}$$

We introduce the *Kalman gain matrix* similar to the classical formulas:

$$\hat{\boldsymbol{K}}_k := \hat{\boldsymbol{P}}_{k|<k}\,\boldsymbol{H}^{\top}\left(\boldsymbol{H}\,\hat{\boldsymbol{P}}_{k|<k}\,\boldsymbol{H}^{\top} + \boldsymbol{R}\right)^{-1}. \tag{11}$$

Note that taking the matrix inverse at this place in eq. (11) is more efficient than in the standard Gaussian formulas (for reference presented in appendix B) if $M \leq N$, which holds for our experiments.

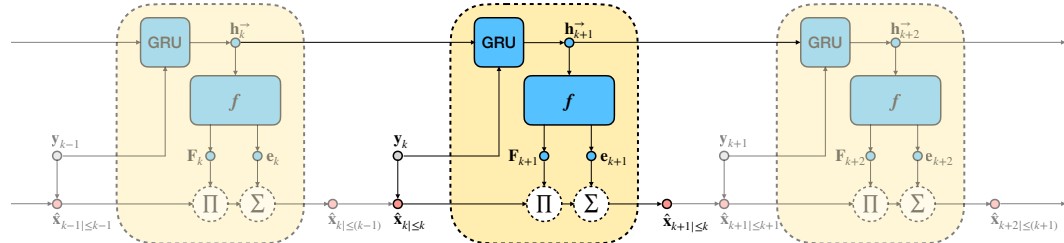

Figure 3: Our recursive model visualized. Data-point $\boldsymbol{y}_k$ is fed, together with hidden state $\boldsymbol{h}_k^{\rightarrow}$, into a GRU unit. The new hidden state $\boldsymbol{h}_{k+1}^{\rightarrow}$ is decoded into multiplicative component $\boldsymbol{F}_{k+1}$ and additive component $\boldsymbol{e}_{k+1}$. Using these, the previous posterior mean $\hat{\boldsymbol{x}}_{k|\leq k}$ is transformed into the prior estimate for time-step $k+1$ $\hat{\boldsymbol{x}}_{k+1\leq k}$. $\boldsymbol{y}_{k+1}$ is used to obtain posterior mean $\hat{\boldsymbol{x}}_{k+1|\leq(k+1)}$.

This completes the recursion: we can use eq. (9) for a new time-step $k+1$ by plugging it back into eq. (6). We have shown how estimating a local linear transition using neural networks in eq. (4) ensures that all the recursive update steps from the Kalman filter analytically hold without specifying and estimating a generative model.

We note that we could also have parameterized

$$p(\boldsymbol{x}_k \mid \boldsymbol{y}_{<k}) = \mathcal{N}\left(\boldsymbol{x}_k \mid \hat{\boldsymbol{x}}_{k|<k}(\boldsymbol{y}_{<k}), \hat{\boldsymbol{P}}_{k|<k}(\boldsymbol{y}_{<k})\right) \tag{12}$$

with $\hat{\boldsymbol{x}}_{k|<k}(\boldsymbol{y}_{<k})$ and $\hat{\boldsymbol{P}}_{k|<k}(\boldsymbol{y}_{<k})$ directly estimated by a neural network. This has the advantage that we do not rely on a local linear transition model. However, it also means that we are estimating $\boldsymbol{x}_k$ without any form of temporal regularization. Additionally, it is harder to incorporate prior knowledge about the transitions maps into such a model. Nonetheless, we detail this parameterization further in the supplementary material (appendix C) and include its performance in our experiments in section 8.

To conclude the section, we like to discuss some of the limitations of the approach. 1. The Gaussianity assumption of eq. (4) ensures but also restricts eq. (8) and eq. (9) to these forms. That is, we make a direct assumption about the form of the posterior $p(\boldsymbol{x}_k \mid \boldsymbol{y}_{\leq k})$. Defending our case, we like to point out that methods such as variational inference (Krishnan et al., 2017) or posterior regularization (Ganchev et al., 2010) also make assumptions (e.g., mean-field Gaussian) about the posterior. 2. Since we did not explicitly specify a factorization of $p(\boldsymbol{x}_{0:K})$, we cannot ensure that the distributions we obtain from the above procedure form a posterior to the ground truth generative process. This does not mean, however, that we cannot perform accurate inference. Arguably, not making explicit assumptions about the generative process is preferred to making wrong assumptions and using those for modeling, which can be the case for variational auto-encoders. 3. The local linearity assumption is justifiable if the length between time-steps is sufficiently small. However, note that the model is more flexible than directly parameterizing eq. (12) (see appendix C) since it *reduces* to that case by putting $\boldsymbol{F}_k := \boldsymbol{0}$ for all $k$.

## 5 FITTING

We have shown in the previous section how parameterization of a local linear transition model leads to recursive estimation of $p(\boldsymbol{x}_k \mid \boldsymbol{y}_{<k})$ and $p(\boldsymbol{x}_k \mid \boldsymbol{y}_{\leq k})$ for all $k$ using classical Bayesian filtering formulas. The inference is only effective if the estimates $\hat{\boldsymbol{F}}_{k|<k}$, $\hat{\boldsymbol{e}}_{k|<k}$ and $\hat{\boldsymbol{Q}}_{k|<k}$ from eq. (4) are accurate. We can use the parameterization $p(\boldsymbol{x}_k \mid \boldsymbol{y}_{<k})$ of eq. (6), the emission model $p(\boldsymbol{y}_k \mid \boldsymbol{x}_k) = \mathcal{N}(\boldsymbol{y}_k \mid \boldsymbol{H}\,\boldsymbol{x}_k, \boldsymbol{R})$ from eq. (3), and the factorization from eq. (1) to see that an analytic form of the log-likelihood of the data emerges:

$$\begin{aligned} p(\boldsymbol{y}_k \mid \boldsymbol{y}_{<k}) &= \int p(\boldsymbol{y}_k \mid \boldsymbol{x}_k)\, p(\boldsymbol{x}_k \mid \boldsymbol{y}_{<k})\, d\boldsymbol{x}_k \\ &= \mathcal{N}\left(\boldsymbol{y}_k \mid \boldsymbol{H}\,\hat{\boldsymbol{x}}_{k|<k}(\boldsymbol{y}_{<k}), \boldsymbol{H}\,\hat{\boldsymbol{P}}_{k|<k}(\boldsymbol{y}_{<k})\,\boldsymbol{H}^\top + \boldsymbol{R}\right), \end{aligned} \tag{13}$$

$$\log p(\boldsymbol{y}_{0:K}) = \sum_{k=0}^{K} \log \mathcal{N} \left( \boldsymbol{y}_k \mid \boldsymbol{H}\, \hat{\boldsymbol{x}}_{k|<k}(\boldsymbol{y}_{<k}), \boldsymbol{H}\, \hat{\boldsymbol{P}}_{k|<k}(\boldsymbol{y}_{<k})\, \boldsymbol{H}^\top + \boldsymbol{R} \right). \tag{14}$$

If we put $\hat{\boldsymbol{y}}_{k|<k} := \boldsymbol{H}\, \hat{\boldsymbol{x}}_{k|<k}(\boldsymbol{y}_{<k})$ and $\hat{\boldsymbol{M}}_{k|<k} := \boldsymbol{H}\, \hat{\boldsymbol{P}}_{k|<k}(\boldsymbol{y}_{<k})\, \boldsymbol{H}^\top + \boldsymbol{R}$, then the maximum-likelihood objective leads to the following loss function, which we can minimize using gradient descent methods w.r.t. all model parameters:

$$\mathcal{L} := \sum_{k=0}^{K} \left[ (\hat{\boldsymbol{y}}_{k|<k} - \boldsymbol{y}_k)^\top \hat{\boldsymbol{M}}_{k|<k}^{-1} (\hat{\boldsymbol{y}}_{k|<k} - \boldsymbol{y}_k) + \log \det \hat{\boldsymbol{M}}_{k|<k} \right]. \tag{15}$$

Note that each term in the sum above represents a one-step-ahead *self-supervised* error term. We thus minimize the prediction residuals $\hat{\boldsymbol{y}}_{k|<k}(\boldsymbol{y}_{<k}) - \boldsymbol{y}_k$ in a norm that is inversely scaled with the above covariance matrix, plus a regularizing determinant term, which prevents the covariance matrix from diverging. The arisen loss function is similar to the `noise2noise` (Lehtinen et al., 2018; Krull et al., 2019; Batson & Royer, 2019; Laine et al., 2019) objective from computer vision literature, combined with a locally linear transition model like Becker et al. (2019). We show in appendix D that this objective will yield correct results (meaning estimating the ground-truth $\boldsymbol{x}_k$) if the noise is independent with $\mathbb{E}[\boldsymbol{y}_k \mid \boldsymbol{H}\boldsymbol{x}_k] = \boldsymbol{H}\boldsymbol{x}_k$. A similar procedure in the causality literature is given by Schölkopf et al. (2016). An algorithmic presentation of performing inference and fitting is presented in appendix E.

Note that after fitting the parameters to the data, eq. (6) can directly be used to do one-step ahead forecasting. Forecasting an arbitrary number of time steps is possible by plugging the new value $\hat{\boldsymbol{x}}_{K+1|K}$ via $\boldsymbol{y}_{K+1} := \boldsymbol{H}\, \hat{\boldsymbol{x}}_{K+1|K}$ back into the recurrent model, and so on. This is not a generative model but merely a convenience that we deemed worth mentioning.

## 6 LINEARIZED SMOOTHING

So far, we have only discussed how to perform filtering. Recall that for smoothing, we are instead interested in the quantity $p(\boldsymbol{x}_k \mid \boldsymbol{y}_{0:K})$. A smoothing strategy highly similar to the methods described earlier can be obtained by explicitly parameterizing such a model, which we detail in the supplementary material. Here, we introduce a *linearized smoothing* procedure. The essential advantage is that no additional model has to be trained, which can be costly. Several algorithms stemming from the Kalman filter literature can be applied, such as the RTS smoother (Rauch et al., 1965) and the two-filter smoother (Kitagawa, 1994). To enable this, we need to assume that the conditional mutual information $I\left(\boldsymbol{x}_{k-1}; \boldsymbol{y}_{k:K} \mid \boldsymbol{x}_k, \boldsymbol{y}_{0:k-1}\right)$ is small for all $k = 1, \dots, K$. In other words, we assume that we approximately have the following conditional independences:

$$\boldsymbol{x}_{k-1} \perp\!\!\!\perp \boldsymbol{y}_{k:K} \mid (\boldsymbol{x}_k, \boldsymbol{y}_{0:k-1}). \tag{16}$$

To explain the motivation for this requirement, consider the model in fig. 2. If the states of $\boldsymbol{y}_{0:k-1}$ and $\boldsymbol{x}_k$ are known, then the additional information that $\boldsymbol{y}_{k:K}$ has about the latent variable $\boldsymbol{x}_{k-1}$ would need to be passed along the unblocked deeper paths like $\boldsymbol{y}_{k+1} \leftarrow \boldsymbol{x}_{k+1} \leftarrow \boldsymbol{e}_{k+1} \leftarrow \boldsymbol{e}_k \leftarrow \boldsymbol{x}_{k-1}$. Then the assumption of small $I(\boldsymbol{x}_{k-1}; \boldsymbol{y}_{k:K} \mid \boldsymbol{x}_k, \boldsymbol{y}_{0:k-1})$ can be interpreted as that the deeper paths transport less information than the lower direct paths. If we consider all edges to the $\boldsymbol{x}_k$'s as linear and the edges to the $\boldsymbol{e}_k$'s as non-linear maps, the above could be interpreted as an information-theoretic version of expressing that the non-linear correction terms are small compared to the linear parts in the functional relations between the variables.

We will now show that under the earlier assumptions and eq. (16) we get a Gaussian approximation: $p(\boldsymbol{x}_k \mid \boldsymbol{y}_{0:K}) \approx \mathcal{N}\left(\boldsymbol{x}_k \mid \hat{\boldsymbol{z}}_k, \hat{\boldsymbol{G}}_k\right)$. We will do backward induction with $\hat{\boldsymbol{z}}_K := \hat{\boldsymbol{x}}_{K|\leq K}$ and $\hat{\boldsymbol{G}}_K := \hat{\boldsymbol{P}}_{K|\leq K}$. To propagate this to previous time steps $k-1$ we use the chain rule:

$$p(\boldsymbol{x}_{k-1} \mid \boldsymbol{y}_{0:K}) = \int p(\boldsymbol{x}_{k-1} \mid \boldsymbol{x}_k, \boldsymbol{y}_{0:K})\, p(\boldsymbol{x}_k \mid \boldsymbol{y}_{0:K})\, d\boldsymbol{x}_k, \tag{17}$$

where the second term is known by backward induction and for the first term we make use of the approximate conditional independence from eq. (16) to get

$$p(\boldsymbol{x}_{k-1} \mid \boldsymbol{x}_k, \boldsymbol{y}_{0:K}) = p(\boldsymbol{x}_{k-1} \mid \boldsymbol{y}_{k:K}, \boldsymbol{x}_k, \boldsymbol{y}_{0:k-1}) \overset{eq.\ (16)}{\approx} p(\boldsymbol{x}_{k-1} \mid \boldsymbol{x}_k, \boldsymbol{y}_{0:k-1}). \tag{18}$$

The latter was shown to be Gaussian in section 4:

$$p(\boldsymbol{x}_{k-1}, \boldsymbol{x}_k \mid \boldsymbol{y}_{0:k-1}) = \mathcal{N}\left(\begin{bmatrix}\boldsymbol{x}_{k-1}\\\boldsymbol{x}_k\end{bmatrix} \mid \begin{bmatrix}\hat{\boldsymbol{x}}_{k-1|\leq k-1}\\\hat{\boldsymbol{x}}_{k|<k}\end{bmatrix}, \begin{bmatrix}\hat{\boldsymbol{P}}_{k-1|\leq k-1} & \hat{\boldsymbol{P}}_{k-1|\leq k-1}\,\hat{\boldsymbol{F}}_{k|<k}^{\top}\\\hat{\boldsymbol{F}}_{k|<k}\,\hat{\boldsymbol{P}}_{k-1|\leq k-1} & \hat{\boldsymbol{P}}_{k|<k}\end{bmatrix}\right).$$
(19)

By use of the usual formulas for Gaussians and the *reverse Kalman gain matrix* $\hat{\boldsymbol{J}}_{k-1|k}$ we arrive at the following update formulas, $k = K, \ldots, 1$, with $\hat{\boldsymbol{z}}_K := \hat{\boldsymbol{x}}_{K|\leq K}$ and $\hat{\boldsymbol{G}}_K := \hat{\boldsymbol{P}}_{K|\leq K}$:

$$\hat{\boldsymbol{J}}_{k-1|k} := \hat{\boldsymbol{P}}_{k-1|\leq k-1}\,\hat{\boldsymbol{F}}_{k|<k}^{\top}\,\hat{\boldsymbol{P}}_{k|<k}^{-1},$$
(20)

$$\hat{\boldsymbol{G}}_{k-1} := \hat{\boldsymbol{P}}_{k-1|\leq k-1} + \hat{\boldsymbol{J}}_{k-1|k}\left(\hat{\boldsymbol{P}}_{k|\leq k} - \hat{\boldsymbol{P}}_{k|<k}\right)\hat{\boldsymbol{J}}_{k-1|k}^{\top},$$
(21)

$$\hat{\boldsymbol{z}}_{k-1} := \hat{\boldsymbol{x}}_{k-1|\leq k-1} + \hat{\boldsymbol{J}}_{k-1|k}\left(\hat{\boldsymbol{z}}_k - \hat{\boldsymbol{x}}_{k|\leq k}\right).$$
(22)

As such, we can perform inference for all $k = 0, \ldots, K$ with $p(\boldsymbol{x}_k \mid \boldsymbol{y}_{0:K}) \approx \mathcal{N}\left(\boldsymbol{x}_k \mid \hat{\boldsymbol{z}}_k, \hat{\boldsymbol{G}}_k\right)$. Algorithmically, the above is presented in appendix E.

## 7 RECURRENT NEURAL NETWORK

Before going into the experiments section, we briefly explain how we specifically estimate the functions $\hat{\boldsymbol{F}}_{k|<k}(\boldsymbol{y}_{<k})$, $\hat{\boldsymbol{e}}_{k|<k}(\boldsymbol{y}_{<k})$ and $\hat{\boldsymbol{Q}}_{k|<k}(\boldsymbol{y}_{<k})$ that parameterize the transition probability $p(\boldsymbol{x}_k \mid \boldsymbol{y}_{<k}, \boldsymbol{x}_{k-1})$ (eq. (4)). The choice of the model here implicitly makes further assumptions about the generative model. If we consider neural networks, the temporal nature of the data suggests recurrent neural networks (Graves et al., 2013), convolutional neural networks (Kalchbrenner et al., 2014), or transformer architectures (Vaswani et al., 2017). Additionally, if the data is image-based, one might further make use of convolutions. For our experiments, we use a Gated Recurrent Unit (GRU) network (Cho et al., 2014), that recursively encodes hidden representations. Therefore, we put

$$\hat{\boldsymbol{Q}}_{k|<k} := \boldsymbol{L}_k \boldsymbol{L}_k^{\top}, \qquad \begin{bmatrix}\hat{\boldsymbol{F}}_k\\\hat{\boldsymbol{e}}_k\\\hat{\boldsymbol{L}}_k\end{bmatrix} := \boldsymbol{f}(\vec{\boldsymbol{h}}_k), \qquad \vec{\boldsymbol{h}}_k := \mathrm{GRU}(\vec{\boldsymbol{y}}_{k-1}, \vec{\boldsymbol{h}}_{k-1}), \qquad (23)$$

where $\hat{\boldsymbol{L}}_k$ is a Cholesky factor and $\boldsymbol{f}$ is a multi-layer perceptron decoder.

## 8 EXPERIMENTS

We perform three experiments, as motivated in section 1. Technical details on the setup of the experiments can be found in the supplementary material (appendix F). We refer to the model detailed in section 4 as the *recursive filter*, as it uses the Bayesian update recursion. For smoothing experiments, we use *recursive smoother*. The model obtained by parameterizing $p(\boldsymbol{x}_k \mid \boldsymbol{y}_{<k})$ directly (eq. (12)) is referred to as the *recurrent filter* or *recurrent smoother*, as it only employs recurrent neural networks (and no Bayesian recursion) to estimate said density directly.

### 8.1 LINEAR DYNAMICS

In the linear Gaussian case, it is known that the Kalman filter will give the optimal solution. Thus, we can get a lower bound on the test loss. In this toy experiment, we simulate particle tracking under linear dynamics and noisy measurements of the location. We use Newtonian physics equations as prior knowledge. We generate trajectories $\mathcal{T} = \{\boldsymbol{x}_{0:K}, \boldsymbol{y}_{0:K}\}$ with $\boldsymbol{x}_k \in \mathbb{R}^6$ and $\boldsymbol{y}_k \in \mathbb{R}^2$ according to the differential equations:

$$\dot{\boldsymbol{x}} = \boldsymbol{A}\boldsymbol{x} = \begin{bmatrix}0 & 1 & 0\\0 & -c & 1\\0 & -\tau c & 0\end{bmatrix}\begin{bmatrix}p\\v\\a\end{bmatrix}. \qquad (24)$$

We obtain sparse, noisy measurements $\boldsymbol{y}_k = \boldsymbol{H}\boldsymbol{x}_k + \boldsymbol{r}$ with $\boldsymbol{r} \sim \mathcal{N}(\boldsymbol{0}, \boldsymbol{R})$. $\boldsymbol{H}$ is a selection matrix that returns a two-dimensional position vector. We run this experiment in a filtering setting, i.e.,

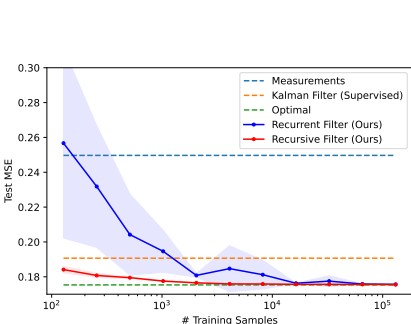
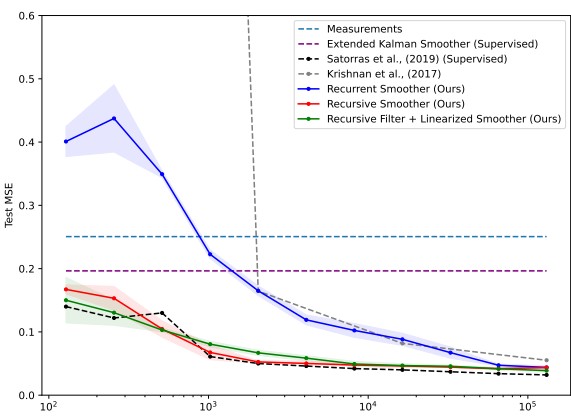

Figure 4: Considers the linear dynamics experiment (section 8.1). The mean squared error on the test set (lower is better) as a function of the number of examples.

Figure 5: Considers the Lorenz experiment (section 8.2). The mean squared error on the test set (lower is better) as a function of the number of examples used for training.

we only use past observations. We compare against (1) the raw, noisy measurements which inherently deviate from the clean measurements, (2) the Kalman filter solution where we optimized the transition covariance matrix using clean data (hence supervised), (3) the optimal solution, which is a Kalman filter with ground truth parameters performing exact inference. To estimate the transition maps for the Kalman filter, we use the standard Taylor series of $e^{A \cdot \Delta t}$ up to the first order. Additionally, we use this expert knowledge as an *inductive bias* for the recursive filter's transition maps.

In fig. 4 we depict the test mean squared error (MSE, lower is better) as a function of the number of training samples. Given enough data, the self-supervised models approximate the optimal solution arbitrarily well. Our recursive model significantly outperforms both the Kalman filter and the inference model in the low-data regime by using incorporated expert knowledge. Additionally, we report that the recursive model's distance to the ground truth latent states is much closer to the optimal solution than both the inference model and Kalman filter. Specifically, we report average mean squared errors of $0.685$ for the inference model, $0.241$ for the Kalman filter, **0.161** for the recursive model compared to $0.135$ for the optimal Kalman filter. Finally, it is worth noting that the recursive model has much less variance as a function of its initialization.

## 8.2 LORENZ EQUATIONS

We simulate a Lorenz system according to

$$\dot{\boldsymbol{x}} = \boldsymbol{A}\boldsymbol{x} = \begin{bmatrix} -\sigma & \sigma & 0 \\ \rho - x_1 & -1 & 0 \\ x_2 & 0 & -\beta \end{bmatrix} \begin{bmatrix} x_1 \\ x_2 \\ x_3 \end{bmatrix}. \tag{25}$$

We have $\boldsymbol{H} = \boldsymbol{I}$, $\boldsymbol{x} \in \mathbb{R}^3$ and $\boldsymbol{y} \in \mathbb{R}^3$. The Lorenz equations model atmospheric convection and form a classic example of chaos. Therefore, performing inference is much more complex than in the linear case. This time, we perform smoothing (see appendix A) and compare against (1) the raw measurements, (2) a supervised Extended Kalman smoother (Ljung, 1979), (3) the variational inference approach of Krishnan et al. (2017), (4) the supervised recursive model of Satorras et al. (2019). Our models include a recurrent smoother, a recursive smoother, and the recursive filter with linearized smoothing (section 6). Transition maps for the Extended Kalman smoother and the recursive models are obtained by taking a second-order Taylor series of $e^{A \cdot \Delta t}$. For the supervised extended Kalman filter, we again optimize its covariance estimate using ground truth data.

In fig. 5 we plot the test mean squared error (MSE, lower is better) as a function of the number of examples available for training. It is clear that our methods approach the ground truth states with

| | Whitenoise | Doing the dishes | Dude miaowing | Exercise Bike | Pink noise | Running tap | Combined |
|---|---|---|---|---|---|---|---|
| Kalman Filter (Supervised) | 0.225 | 0.230 | 0.232 | 0.237 | 0.235 | 0.230 | 0.227 |
| Noise2Noise (Lehtinen et al., 2018) | 0.327 | 0.399 | 0.448 | 0.430 | 0.440 | 0.383 | 0.526 |
| SIN (Krishnan et al., 2017) | 0.297 | 0.373 | 0.352 | 0.348 | 0.377 | 0.342 | 0.343 |
| Recurrent Filter (Ours) | **0.102** | 0.207 | 0.213 | 0.200 | 0.234 | 0.175 | 0.181 |
| Recursive Filter (Ours) | 0.107 | 0.206 | **0.213** | 0.198 | 0.232 | 0.166 | **0.175** |
| Recursive Filter + RTS Smoother (Ours) | **0.100** | **0.204** | 0.215 | **0.197** | **0.231** | **0.166** | 0.176 |
| RKN (Becker et al., 2019, Supervised) | 0.121 | **0.127** | **0.109** | **0.105** | **0.085** | **0.121** | **0.125** |

Table 2: Considers the audio denoising experiment (section 8.3). Test mean squared error (MSE, lower is better) per model and noise subset. Blue numbers indicate second-best performing models.

more data. This is in contrast to the Extended Kalman smoother, which barely outperforms the noisy measurements. We also see that the recursive models significantly outperform the recurrent model in the low-data regime. The recursive filter with linearized smoothing performs comparably to the other models and even better in low-data regimes. We hypothesize that this is because the required assumption for the linearized smoother holds (section 6) and regularizes the model. The variational method of Krishnan et al. (2017) performs poorly in low-data regimes. Most notably, the supervised method of Satorras et al. (2019) outperforms our models only slightly.

## 8.3 AUDIO DENOISING

Next, we test the model on non-fabricated data with less ideal noise characteristics. Specifically, we use the `SpeechCommands` spoken audio dataset (Warden, 2018). Performing inference on spoken audio is challenging, as it arguably requires understanding natural language. To this end, recent progress on synthesizing raw audio has been made (Lakhotia et al., 2021). However, this requires scaling to much larger and more sophisticated neural networks than presented here, which we deem out the current work's scope. Therefore, we take a subset of the entire dataset, using audio from the classes "tree", "six", "eight", "yes", and "cat". We overlay these clean audio sequences $\{x_{0:K}^{(1)}, \ldots, x_{0:K}^{(N)}\}$ with various noise classes that the dataset provides. That is, for every noise class $C$ we obtain a set of noisy sequences $\{y_{0:K}^{(1)}, \ldots, y_{0:K}^{(N)}\}_C$. We also consider a "combined" class in which we sample from the union of the noise sets. The task is to denoise the audio without having access to clean data. We evaluate the models on non-silent parts of the audio, as performance on those sections is the most interesting. Notably, none of these noise classes is Gaussian distributed.

We show the mean squared error on the test set of all models per noise class in table 2. Our models outperform the Kalman filter, Noise2Noise (Lehtinen et al., 2018), and SIN (Krishnan et al., 2017) unsupervised baselines. We suspect that the relatively poor performance of SIN is due to its generative Markov assumption, regularizing the model too strongly. The poor performance of Noise2Noise is due to the fact that it does not use the current measurement $y_k$ to infer $x_k$. Like before, note that the Kalman filter is "supervised" as we optimized its covariance matrix using clean data. The supervised RKN (Becker et al., 2019) outperforms our models on most noise classes, but notably not on white noise. Most of these noise classes have temporal structure, making them predictable from past data. This is confirmed by observing these mean squared error values over the course of training. Initially, the values were better than reported in table 2, but the model increasingly fits the noise over time. Thus, although two of the main assumptions about the model (independent Gaussian noise) are violated, we still can denoise effectively. Since the RKN's targets are denoised (hence "supervised"), it does not have this problem. In practice, obtaining clean data can be challenging.

## 9 CONCLUSION

We presented an advantageous parameterization of an inference procedure for nonlinear state-space models with potentially higher-order latent Markov chains. The inference distribution is split into linear and nonlinear parts, allowing for a recursion akin to the Kalman filter and smoother algorithms. Optimization is performed directly using a maximum-likelihood objective that backpropagates through these recursions. Smoothing can be performed similarly, but we additionally proposed linearized smoothing that can directly be applied to the filtering distributions. Our model is simple and builds on established methods from signal processing. Despite this, results showed that it can perform better or on par with fully supervised or variational inference methods.

## 10 ETHICS STATEMENT

The paper presents a simple method to perform inference using noisy sequential data. Applications can be found throughout society, e.g., tracking particles, denoising images or audio, or estimating system states. While many such examples are for good, there are applications with ethically debatable motivations. Among these could be tracking humans or denoising purposefully corrupted data (e.g., to hide one's identity).

## 11 REPRODUCIBILITY STATEMENT

We are in the process of releasing code for the current work. For clarity and reproducibility, the presented methods are available as algorithms in the supplementary material. Furthermore, we made explicit wherever we needed to make an approximation or an assumption.

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

## A    PARAMETERIZED SMOOTHING

In the main body of the paper, we showed how to parameterize the model for recursive estimation of $p(\boldsymbol{x}_k \mid \boldsymbol{y}_{\leq k})$, $k = 0, \ldots, K$. Additionally, we provided a *linearized smoothing* procedure that yields $p(\boldsymbol{x}_k \mid \boldsymbol{y}_{1:K})$. The disadvantage clearly is the linearization. Here, we show how we can recursively estimate $p(\boldsymbol{x}_k \mid \boldsymbol{y}_{1:K})$ in a similar sense to the filtering setting.

First, we put

$$p(\boldsymbol{x}_k \mid \boldsymbol{x}_{k-1}, \boldsymbol{y}_{-k}) = \mathcal{N}\left(\boldsymbol{x}_k \mid \hat{\boldsymbol{F}}_{k|-k}\,\boldsymbol{x}_{k-1} + \hat{e}_{k|-k}, \hat{\boldsymbol{Q}}_{k|-k}\right), \tag{26}$$

where $\hat{\boldsymbol{F}}_{k|-k}(\boldsymbol{y}_{-k})$ and $\hat{e}_{k|-k}(\boldsymbol{y}_{-k})$ and $\hat{\boldsymbol{Q}}_{k|-k}(\boldsymbol{y}_{-k})$ are two-sided recurrent neural network outputs similar to section 7. We compute the distribution of interest as follows:

$$p(\boldsymbol{x}_k \mid \boldsymbol{y}_{0:K}) = \frac{p(\boldsymbol{y}_k \mid \boldsymbol{x}_k)p(\boldsymbol{x}_k \mid \boldsymbol{y}_{-k})}{p(\boldsymbol{y}_k \mid \boldsymbol{y}_{-k})} \tag{27}$$

$$\propto p(\boldsymbol{y}_k \mid \boldsymbol{x}_k) \int p(\boldsymbol{x}_k, \boldsymbol{x}_{k-1} \mid \boldsymbol{y}_{-k}) d\boldsymbol{x}_{k-1} \tag{28}$$

$$= p(\boldsymbol{y}_k \mid \boldsymbol{x}_k) \int p(\boldsymbol{x}_k \mid \boldsymbol{x}_{k-1}, \boldsymbol{y}_{-k})p(\boldsymbol{x}_{k-1} \mid \boldsymbol{y}_{-k}) d\boldsymbol{x}_{k-1} \tag{29}$$

$$\stackrel{\boldsymbol{x}_{k-1} \perp\!\!\!\perp \boldsymbol{y}_k \mid \boldsymbol{y}_{-k}}{\approx} p(\boldsymbol{y}_k \mid \boldsymbol{x}_k) \int p(\boldsymbol{x}_k \mid \boldsymbol{x}_{k-1}, \boldsymbol{y}_{-k})p(\boldsymbol{x}_{k-1} \mid \boldsymbol{y}_{0:K}) d\boldsymbol{x}_{k-1} \tag{30}$$

where we make the approximation to compute eq. (27) efficiently and recursively. It is justified if $I(\boldsymbol{x}_{k-1}; \boldsymbol{y}_k \mid \boldsymbol{y}_{-k}) < \epsilon$ for small $\epsilon$. That is, the additional information that $\boldsymbol{y}_k$ conveys about $\boldsymbol{x}_{k-1}$ is marginal if we have all other data. Let

$$p(\boldsymbol{x}_{k-1} \mid \boldsymbol{y}_{0:K}) := \mathcal{N}\left(\boldsymbol{x}_{k-1} \mid \hat{\boldsymbol{x}}_{k-1|0:K}, \hat{\boldsymbol{P}}_{k-1|0:K}\right) \tag{31}$$

be previous time-step's posterior. Then put

$$p(\boldsymbol{x}_k \mid \boldsymbol{x}_{k-1}, \boldsymbol{y}_{-k}) := \mathcal{N}(\boldsymbol{x} \mid \hat{\boldsymbol{F}}_{k|-k}\boldsymbol{x}_{k-1} + \hat{e}_{k|-k}, \hat{\boldsymbol{Q}}_{k|-k}). \tag{32}$$

where we left out the arguments for the following quantities estimated by an RNN.

$$\begin{bmatrix} \hat{\boldsymbol{F}}_{k|-k}(\boldsymbol{y}_{-k}) \\ \hat{e}_{k|-k}(\boldsymbol{y}_{-k}) \\ \hat{\boldsymbol{Q}}_{k|-k}(\boldsymbol{y}_{-k}) \end{bmatrix} := \boldsymbol{f}(\boldsymbol{h}_k^{\rightarrow}, \boldsymbol{h}_k^{\leftarrow}) \qquad\qquad \begin{bmatrix} \boldsymbol{h}_k^{\rightarrow} \\ \boldsymbol{h}_k^{\leftarrow} \end{bmatrix} := \begin{bmatrix} \mathrm{GRU}(\boldsymbol{h}_{k-1}^{\rightarrow}, \boldsymbol{y}_{k-1}) \\ \mathrm{GRU}(\boldsymbol{h}_{k+1}^{\leftarrow}, \boldsymbol{y}_{k+1}) \end{bmatrix} \tag{33}$$

Applying the integral in eq. (30) we get

$$\int p(\boldsymbol{x}_k \mid \boldsymbol{x}_{k-1}, \boldsymbol{y}_{-k}) p(\boldsymbol{x}_{k-1} \mid \boldsymbol{y}_{0:K}) d\boldsymbol{x}_{k-1} = \mathcal{N}\left(\boldsymbol{x}_k \mid \hat{\boldsymbol{x}}_{k|-k}, \hat{\boldsymbol{P}}_{k|-k}\right) \tag{34}$$

where we put

$$\hat{\boldsymbol{x}}_{k|-k} := \hat{\boldsymbol{F}}_{k|-k} \hat{\boldsymbol{x}}_{k-1|0:K} + \hat{\boldsymbol{e}}_{k|-k} \qquad \hat{\boldsymbol{P}}_{k|-k} := \hat{\boldsymbol{F}}_{k|-k} \hat{\boldsymbol{P}}_{k-1|0:K} \hat{\boldsymbol{F}}_{k|-k}^\top + \hat{\boldsymbol{Q}}_{k|-k} \tag{35}$$

A data likelihood can be computed as follows

$$p(\boldsymbol{y}_k \mid \boldsymbol{y}_{-k}) = \int p(\boldsymbol{y}_k \mid \boldsymbol{x}_k) p(\boldsymbol{x}_k \mid \boldsymbol{y}_{-k}) d\boldsymbol{x}_k \tag{36}$$

$$= \int p(\boldsymbol{y}_k \mid \boldsymbol{x}_k) \int p(\boldsymbol{x}_{k-1}, \boldsymbol{x}_k \mid \boldsymbol{y}_{-k}) d\boldsymbol{x}_k \boldsymbol{x}_{k-1} \tag{37}$$

$$= \int p(\boldsymbol{y}_k \mid \boldsymbol{x}_k) \int p(\boldsymbol{x}_k \mid \boldsymbol{x}_{k-1}, \boldsymbol{y}_{-k}) p(\boldsymbol{x}_{k-1} \mid \boldsymbol{y}_{-k}) d\boldsymbol{x}_k \boldsymbol{x}_{k-1} \tag{38}$$

$$\stackrel{\boldsymbol{x}_{k-1} \perp\!\!\!\perp \boldsymbol{y}_k \mid \boldsymbol{y}_{-k}}{\approx} \int p(\boldsymbol{y}_k \mid \boldsymbol{x}_k) \int p(\boldsymbol{x}_k \mid \boldsymbol{x}_{k-1}, \boldsymbol{y}_{-k}) p(\boldsymbol{x}_{k-1} \mid \boldsymbol{y}_{0:K}) d\boldsymbol{x}_k \boldsymbol{x}_{k-1} \tag{39}$$

where we made the same approximation eq. (30) as before. It evaluates to

$$\int p(\boldsymbol{y}_k \mid \boldsymbol{x}_k) p(\boldsymbol{x}_k \mid \boldsymbol{y}_{-k}) d\boldsymbol{x}_k = \mathcal{N}\left(\boldsymbol{y}_k \mid \hat{\boldsymbol{y}}_{k|-k}, \hat{\boldsymbol{M}}_{k|-k}\right) \tag{40}$$

with

$$\hat{\boldsymbol{y}}_{k|-k} := \boldsymbol{H}\hat{\boldsymbol{x}}_{k|-k} \qquad\qquad \hat{\boldsymbol{M}}_{k|-k} := \boldsymbol{H}\hat{\boldsymbol{P}}_{k|-k}\boldsymbol{H}^\top + \boldsymbol{R} \tag{41}$$

For fitting to the data we now would use a maximum-pseudo-likelihood (Gourieroux et al., 1984) approach by maximizing:

$$\sum_{k=0}^{K} \log p(\boldsymbol{y}_k \mid \boldsymbol{y}_{-k}) = \sum_{k=0}^{K} \log \mathcal{N}\left(\boldsymbol{y}_k \mid \boldsymbol{H}\,\hat{\boldsymbol{x}}_{k|-k}(\boldsymbol{y}_{-k}), \boldsymbol{H}\,\hat{\boldsymbol{P}}_{k|-k}(\boldsymbol{y}_{-k})\,\boldsymbol{H}^\top + \boldsymbol{R}\right), \tag{42}$$

leading to minimizing the following self-supervised loss function:

$$\mathcal{L} := \sum_{k=0}^{K} \left[ (\hat{\boldsymbol{y}}_{k|-k} - \boldsymbol{y}_k)^\top \hat{\boldsymbol{M}}_{k|-k}^{-1} (\hat{\boldsymbol{y}}_{k|-k} - \boldsymbol{y}_k) + \log \det \hat{\boldsymbol{M}}_{k|-k} \right], \tag{43}$$

where $\hat{\boldsymbol{y}}_{k|-k} := \boldsymbol{H}\,\hat{\boldsymbol{x}}_{k|-k}(\boldsymbol{y}_{-k})$ and $\hat{\boldsymbol{M}}_{k|-k} := \boldsymbol{H}\,\hat{\boldsymbol{P}}_{k|-k}(\boldsymbol{y}_{-k})\,\boldsymbol{H}^\top + \boldsymbol{R}$.

## B  GAUSSIAN CONDITIONING FORMULAS

Since many of the calculations used in this work are based on the Gaussian conditioning formulas, we provide them here. If

$$p(\boldsymbol{x}) = \mathcal{N}(\boldsymbol{x} \mid \boldsymbol{\mu}, \boldsymbol{P}), \tag{44}$$
$$p(\boldsymbol{y} \mid \boldsymbol{x}) = \mathcal{N}(\boldsymbol{y} \mid \boldsymbol{H}\boldsymbol{x} + \boldsymbol{b}, \boldsymbol{R}), \tag{45}$$

then

$$p(\boldsymbol{y}) = \mathcal{N}(\boldsymbol{y} \mid \boldsymbol{H}\boldsymbol{\mu} + \boldsymbol{b}, \boldsymbol{R} + \boldsymbol{H}\boldsymbol{P}\boldsymbol{H}^\top), \tag{46}$$
$$p(\boldsymbol{x} \mid \boldsymbol{y}) = \mathcal{N}\left(\boldsymbol{x} \mid \boldsymbol{\Sigma}\left[\boldsymbol{H}^\top \boldsymbol{R}^{-1}(\boldsymbol{y} - \boldsymbol{b}) + \boldsymbol{P}^{-1}\boldsymbol{\mu}\right], \boldsymbol{\Sigma}\right), \tag{47}$$

with

$$\boldsymbol{\Sigma} = (\boldsymbol{P}^{-1} + \boldsymbol{H}^\top \boldsymbol{R}^{-1} \boldsymbol{H})^{-1}. \tag{48}$$

## C  DIRECT PARAMETERIZATION OF $p(\boldsymbol{x} \mid \boldsymbol{y}_{<k})$

We show here how to directly parameterize $p(\boldsymbol{x} \mid \boldsymbol{y}_{<k})$ and $p(\boldsymbol{x} \mid \boldsymbol{y}_{-k})$. This parameterization is referred to as the *recurrent model* in our experiments. The procedure is rather straightforward. For filtering, we put $p(\boldsymbol{x}_k \mid \boldsymbol{y}_{<k}) = \mathcal{N}\left(\boldsymbol{x}_k \mid \hat{\boldsymbol{x}}_{k|-k}(\boldsymbol{y}_{<k}), \hat{\boldsymbol{P}}_{k|<k}(\boldsymbol{y}_{<k})\right)$. We model:

$$\hat{\boldsymbol{x}}_{k|<k} := \boldsymbol{e}_k, \qquad \hat{\boldsymbol{P}}_{k|<k} := \boldsymbol{L}_k \boldsymbol{L}_k^\top, \qquad \begin{bmatrix} \boldsymbol{e}_k \\ \boldsymbol{L}_k \end{bmatrix} := \boldsymbol{f}(\boldsymbol{h}_k^\rightarrow), \tag{49}$$

where $\boldsymbol{L}_k$ is a cholesky factor and $\boldsymbol{f}$ is a multi-layer perceptron. The argument $\boldsymbol{h}_k^\rightarrow$ is recursively given by

$$\boldsymbol{h}_k^\rightarrow := \mathrm{GRU}(\boldsymbol{y}_{k-1}, \boldsymbol{h}_{k-1}^\rightarrow), \tag{50}$$

where we employ a Gated Recurrent Unit (GRU) network (Cho et al., 2014). Note that this parameterization is equivalent to the model described in the main paper with $\boldsymbol{F}_k := \boldsymbol{0}$.

For smoothing, $p(\boldsymbol{x}_k \mid \boldsymbol{y}_{-k}) = \mathcal{N}\left(\boldsymbol{x}_k \mid \hat{\boldsymbol{x}}_{k|-k}(\boldsymbol{y}_{-k}), \hat{\boldsymbol{P}}_{k|-k}(\boldsymbol{y}_{-k})\right)$.

$$\hat{\boldsymbol{x}}_{k|-k} := \boldsymbol{e}_k, \qquad \hat{\boldsymbol{P}}_{k|-k} := \boldsymbol{L}_k \boldsymbol{L}_k^\top, \qquad \begin{bmatrix} \boldsymbol{e}_k \\ \boldsymbol{L}_k \end{bmatrix} := \boldsymbol{f}(\boldsymbol{h}_k^\rightarrow, \boldsymbol{h}_k^\leftarrow), \tag{51}$$

where $\boldsymbol{L}_k$ is a cholesky factor and $\boldsymbol{f}$ is a multi-layer perceptron.

$$\boldsymbol{h}_k^\rightarrow := \mathrm{GRU}(\boldsymbol{y}_{k-1}, \boldsymbol{h}_{k-1}^\rightarrow), \qquad \boldsymbol{h}_k^\leftarrow := \mathrm{GRU}(\boldsymbol{y}_{k+1}, \boldsymbol{h}_{k+1}^\leftarrow). \tag{52}$$

Once $p(\boldsymbol{x} \mid \boldsymbol{y}_{<k})$ (filtering) or $p(\boldsymbol{x} \mid \boldsymbol{y}_{-k})$ (smoothing) is obtained, all the procedures for inference and optimization described in the main paper and appendix A remain the same.

## D  BIAS-VARIANCE-NOISE DECOMPOSITION OF THE SELF-SUPERVISED GENERALIZATION ERROR

Any estimate $\hat{\boldsymbol{x}}_k = \hat{\boldsymbol{x}}_k(\boldsymbol{y}_{-k})$ for $\boldsymbol{x}_k$ that is not dependent on $\boldsymbol{y}_k$ will give us a bias-variance-noise decomposition of the generalization error. Note that this setting covers both the filtering and smoothing case. Define "optimal model" $\hat{\boldsymbol{y}}_k^* := \mathbb{E}\left[\boldsymbol{y}_k \mid \boldsymbol{y}_{-k}\right]$, then under the specified generative model (section 3) we have

$$\mathbb{E}\left[\|\hat{\boldsymbol{y}}_k - \boldsymbol{y}_k\|_2^2 \mid \boldsymbol{y}_{-k}\right] = \mathbb{E}\left[\|\hat{\boldsymbol{y}}_k - \hat{\boldsymbol{y}}_k^* + \hat{\boldsymbol{y}}_k^* - \boldsymbol{y}_k\|_2^2 \mid \boldsymbol{y}_{-k}\right] \tag{53}$$

$$= \mathbb{E}\left[\|\hat{\boldsymbol{y}}_k - \hat{\boldsymbol{y}}_k^*\|_2^2 \mid \boldsymbol{y}_{-k}\right] + \mathrm{Var}\left[\boldsymbol{y}_k \mid \boldsymbol{y}_{-k}\right] + 2\mathbb{E}\left[(\hat{\boldsymbol{y}}_k - \hat{\boldsymbol{y}}_k^*)^\top (\hat{\boldsymbol{y}}_k^* - \boldsymbol{y}_k) \mid \boldsymbol{y}_{-k}\right] \tag{54}$$

$$= \|\hat{\boldsymbol{y}}_k - \hat{\boldsymbol{y}}_k^*\|_2^2 + \mathrm{Var}\left[\boldsymbol{y}_k \mid \boldsymbol{y}_{-k}\right] + 2(\hat{\boldsymbol{y}}_k - \hat{\boldsymbol{y}}_k^*)^\top \underbrace{\mathbb{E}\left[(\hat{\boldsymbol{y}}_k^* - \boldsymbol{y}_k) \mid \boldsymbol{y}_{-k}\right]}_{0} \tag{55}$$

$$\mathrm{Var}\left[\boldsymbol{y}_k \mid \boldsymbol{y}_{-k}\right] = \mathbb{E}\left[\|\hat{\boldsymbol{y}}_k^* - \boldsymbol{H}\boldsymbol{x}_k - \boldsymbol{n}_k\|_2^2 \mid \boldsymbol{y}_{-k}\right] \tag{56}$$

$$= \mathbb{E}\left[\|\hat{\boldsymbol{y}}_k^* - \boldsymbol{H}\boldsymbol{x}_k\|_2^2 \mid \boldsymbol{y}_{-k}\right] + \mathbb{E}\left[\|\boldsymbol{n}_k\|_2^2 \mid \boldsymbol{y}_{-k}\right] + 2\underbrace{\mathbb{E}\left[(\boldsymbol{H}\boldsymbol{x}_k - \hat{\boldsymbol{y}}_k^*)^\top \boldsymbol{n}_k \mid \boldsymbol{y}_{-k}\right]}_{0} \tag{57}$$

$$\stackrel{\boldsymbol{x}_k, \boldsymbol{y}_{-k} \perp\!\!\!\perp \boldsymbol{n}_k}{=} \mathbb{E}\left[\|\hat{\boldsymbol{y}}_k^* - \boldsymbol{H}\boldsymbol{x}_k\|_2^2 \mid \boldsymbol{y}_{-k}\right] + \mathrm{tr}(\boldsymbol{R}) \tag{58}$$

Thus,

$$\mathbb{E}\left[\|\hat{\boldsymbol{y}}_k - \boldsymbol{y}_k\|_2^2 \mid \boldsymbol{y}_{-k}\right] = \|\hat{\boldsymbol{y}}_k - \hat{\boldsymbol{y}}_k^*\|_2^2 + \mathbb{E}\left[\|\hat{\boldsymbol{y}}_k^* - \boldsymbol{H}\boldsymbol{x}_k\|_2^2 \mid \boldsymbol{y}_{-k}\right] + \mathrm{tr}(\boldsymbol{R}) \tag{59}$$

Note that $\hat{\boldsymbol{y}}_k^* = \mathbb{E}\left[\boldsymbol{H}\boldsymbol{x}_k \mid \boldsymbol{y}_{-k}\right] = \boldsymbol{H}\mathbb{E}\left[\boldsymbol{x}_k \mid \boldsymbol{y}_{-k}\right]$. Then, define our model $\hat{\boldsymbol{y}}_k := \boldsymbol{H}\hat{\boldsymbol{x}}_k$. The *reducible* part of the error becomes

$$\|\hat{\boldsymbol{y}}_k^* - \hat{\boldsymbol{y}}_k\|_2^2 = \|\boldsymbol{H}\left(\mathbb{E}[\boldsymbol{x}_k \mid \boldsymbol{y}_{-k}] - \hat{\boldsymbol{x}}_k\right)\|_2^2 \tag{60}$$

For this reason, the model output $\hat{\boldsymbol{x}}_k$ approaches the optimal model under minimization of the self-supervised error (perturbed by $\boldsymbol{H}$).

Additionally, the model $\hat{\boldsymbol{y}}_k$ approaches $\boldsymbol{H}\boldsymbol{x}_k$ (the uncorrupted measurement) under this criterion.

$$\mathbb{E}\left[\|\hat{\boldsymbol{y}}_k - \boldsymbol{y}_k\|^2\right] = \mathbb{E}\left[\|\hat{\boldsymbol{y}}_k - \boldsymbol{H}\boldsymbol{x}_k + \boldsymbol{H}\boldsymbol{x}_k - \boldsymbol{y}_k\|^2\right] \tag{61}$$

$$= \mathbb{E}\left[\|\hat{\boldsymbol{y}}_k - \boldsymbol{H}\boldsymbol{x}_k\|^2\right] + \mathbb{E}\left[\|\boldsymbol{H}\boldsymbol{x}_k - \boldsymbol{y}_k\|^2\right] + 2\,\mathbb{E}\left[(\hat{\boldsymbol{y}}_k - \boldsymbol{H}\boldsymbol{x}_k)^\top (\boldsymbol{H}\boldsymbol{x}_k - \boldsymbol{y}_k)\right] \tag{62}$$

$$= \mathbb{E}\left[\|\hat{\boldsymbol{y}}_k - \boldsymbol{H}\boldsymbol{x}_k\|^2\right] + \mathbb{E}\left[\|\boldsymbol{H}\boldsymbol{x}_k - \boldsymbol{y}_k\|^2\right] + 2\,\mathbb{E}\left[(\hat{\boldsymbol{y}}_k - \boldsymbol{H}\boldsymbol{x}_k)\right]^\top \underbrace{\mathbb{E}\left[\boldsymbol{H}\boldsymbol{x}_k - \boldsymbol{y}_k\right]}_{0} \tag{63}$$

$$= \mathbb{E}\left[\|\hat{\boldsymbol{y}}_k - \boldsymbol{H}\boldsymbol{x}_k\|^2\right] + \mathrm{tr}(\boldsymbol{R}) + \underbrace{\mathbb{E}\left[\|\boldsymbol{H}\boldsymbol{x}_k - \boldsymbol{y}_k\|\right]^2}_{0} \tag{64}$$

If we then take $\hat{\boldsymbol{y}}_k := \boldsymbol{H}\hat{\boldsymbol{x}}_k$ then the reducible part of the error approximates the true $\boldsymbol{x}_k$ (perturbed by $\boldsymbol{H}$).

$$\mathbb{E}\left[\|\hat{\boldsymbol{y}}_k - \boldsymbol{H}\boldsymbol{x}_k\|^2\right] = \mathbb{E}\left[\|\boldsymbol{H}(\hat{\boldsymbol{x}}_k - \boldsymbol{x}_k)\|^2\right] \tag{65}$$

# E   ALGORITHMS

---

**Algorithm 1:** Recursive Filter (Inference)

---

**input** : Data (time-series) $\boldsymbol{y}_{0:K} = (\boldsymbol{y}_0, \ldots, \boldsymbol{y}_K)$, emission matrices $\boldsymbol{H}$ and $\boldsymbol{R}$, parameters $\phi$.
**output:**
    1. For training: Loss value $\mathcal{L}_K$ and its gradient $\nabla_\phi \mathcal{L}_K$.

    2. For inference: $\hat{\boldsymbol{x}}_{k|\leq k}$ and $\hat{\boldsymbol{P}}_{k|\leq k}$ for $k$ in $0, \ldots, K$. Inference is done via:
$$p(\boldsymbol{x}_k \mid \boldsymbol{y}_{\leq k}) = \mathcal{N}\left(\boldsymbol{x}_k \mid \hat{\boldsymbol{x}}_{k|\leq k}, \hat{\boldsymbol{P}}_{k|\leq k}\right).$$

    3. For linearized smoothing (section 6): $\hat{\boldsymbol{F}}_{k|<k}$, $\hat{\boldsymbol{P}}_{k|<k}$, $\hat{\boldsymbol{x}}_{k|\leq k}$ and $\hat{\boldsymbol{P}}_{k|\leq k}$, for $k$ in $0, \ldots, K$.

    4. For forecasting: $\hat{\boldsymbol{x}}_{K+1|<(K+1)}$ and $\hat{\boldsymbol{P}}_{K+1|<(K+1)}$. Forecasting is done via:
$$p(\boldsymbol{x}_{K+1} \mid \boldsymbol{y}_{0:K}) = \mathcal{N}\left(\boldsymbol{x}_{K+1} \mid \hat{\boldsymbol{x}}_{K+1|<(K+1)}, \hat{\boldsymbol{P}}_{K+1|<(K+1)}\right).$$

---

$\boldsymbol{h}_0 := \boldsymbol{0}$
$\mathcal{L}_{-1} := 0$
$\hat{\boldsymbol{P}}_{0|<0} := \hat{\boldsymbol{Q}}_0$
$\hat{\boldsymbol{x}}_{0|<0} := \hat{\boldsymbol{e}}_0$
**for** $k$ *in* $0, \ldots, K$ **do**

$$\hat{\boldsymbol{B}}_{k|<k} := \left(\boldsymbol{H}\,\hat{\boldsymbol{P}}_{k|<k}\,\boldsymbol{H}^\top + \boldsymbol{R}\right)^{-1}$$

$$\hat{\boldsymbol{y}}_{k|<k} := \boldsymbol{H}\,\hat{\boldsymbol{x}}_{k|<k}$$

$$\mathcal{L}_k := \mathcal{L}_{k-1} + (\boldsymbol{y}_k - \hat{\boldsymbol{y}}_{k|<k})^\top \hat{\boldsymbol{B}}_{k|<k} (\boldsymbol{y}_k - \hat{\boldsymbol{y}}_{k|<k}) - \log \det \hat{\boldsymbol{B}}_{k|<k}$$

$$\hat{\boldsymbol{K}}_k := \hat{\boldsymbol{P}}_{k|<k}\,\boldsymbol{H}^\top \hat{\boldsymbol{B}}_{k|<k}$$

$$\hat{\boldsymbol{P}}_{k|\leq k} := \hat{\boldsymbol{P}}_{k|<k} - \hat{\boldsymbol{K}}_k\,\boldsymbol{H}\,\hat{\boldsymbol{P}}_{k|<k}$$

$$\hat{\boldsymbol{x}}_{k|\leq k} := \hat{\boldsymbol{x}}_{k|<k} + \hat{\boldsymbol{K}}_k (\boldsymbol{y}_k - \hat{\boldsymbol{y}}_{k|<k})$$

$$\boldsymbol{h}_{k+1} := \mathrm{GRU}_\phi(\boldsymbol{h}_k, \boldsymbol{y}_k)$$

$$\begin{bmatrix} \hat{\boldsymbol{e}}_{k+1|<(k+1)} \\ \hat{\boldsymbol{F}}_{k+1|<(k+1)} \\ \hat{\boldsymbol{L}}_{k+1|<(k+1)} \end{bmatrix} := \boldsymbol{f}_\phi(\boldsymbol{h}_{k+1})$$

$$\hat{\boldsymbol{Q}}_{k+1|<(k+1)} := \hat{\boldsymbol{L}}_{k+1|<(k+1)}\,\hat{\boldsymbol{L}}^\top_{k+1|<(k+1)}$$

$$\hat{\boldsymbol{P}}_{k+1|<(k+1)} := \hat{\boldsymbol{F}}_{k+1|<(k+1)}\,\hat{\boldsymbol{P}}_{k|\leq k}\,\hat{\boldsymbol{F}}^\top_{k+1|<(k+1)} + \hat{\boldsymbol{Q}}_{k+1|<(k+1)}$$

$$\hat{\boldsymbol{x}}_{k+1|<(k+1)} := \hat{\boldsymbol{F}}_{k+1|<(k+1)}\,\hat{\boldsymbol{x}}_{k|\leq k} + \hat{\boldsymbol{e}}_{k+1|<(k+1)}$$

**end**
For the training case we use backpropagation through the above loop to compute $\nabla_\phi \mathcal{L}_K$.

---

**Algorithm 2:** Recursive Filter (Training)

---

**input** : Data (time-series) $\boldsymbol{y}_{0:K} = (\boldsymbol{y}_0, \ldots, \boldsymbol{y}_K)$, , emission matrices $\boldsymbol{H}$ and $\boldsymbol{R}$, initialized
           parameters $\phi_0$, number of training rounds $I$.
**output:** Model parameters $\phi^*$ for inference at test-time.
**for** $i$ *in* $0, \ldots, I$ **do**
     Obtain $\mathcal{L}_K^{(i)}$ and $\nabla_\phi \mathcal{L}_K^{(i)}$ from algorithm 1.
     Run preferred optimizer step to update parameters $\phi$ with $\nabla_\phi \mathcal{L}_K^{(i)}$ (and $\mathcal{L}_K^{(i)}$).
**end**

---

---

**Algorithm 3:** Parameterized Recursive Smoother (Inference)

---

**input** : Data (time-series) $\boldsymbol{y}_{0:K} = (\boldsymbol{y}_0, \ldots, \boldsymbol{y}_K)$, emission matrices $\boldsymbol{H}$ and $\boldsymbol{R}$, initialized parameters $\phi_0$

**output:**

1. Loss value $\mathcal{L}_K$ and its gradient w.r.t. all model parameters $\nabla_\phi(\mathcal{L}_K)$ for training.

2. For all $k$ in $0, \ldots, K$: $\hat{\boldsymbol{x}}_{k|0:K}, \hat{\boldsymbol{P}}_{k|0:K}$. These can be used for inference through
$$p(\boldsymbol{x}_k \mid \boldsymbol{y}_{0:K}) = \mathcal{N}\left(\hat{\boldsymbol{x}}_{k|0:K}, \hat{\boldsymbol{P}}_{k|0:K}\right).$$

---

$\boldsymbol{h}_0^{\rightarrow} := \boldsymbol{0}$
$\boldsymbol{h}_{K+1}^{\leftarrow} := \boldsymbol{0}$
$\mathcal{L}_{-1}^{(i)} := 0$
$\hat{\boldsymbol{P}}_{0|-0} := \hat{\boldsymbol{Q}}_0$
$\hat{\boldsymbol{x}}_{0|-0} := \hat{\boldsymbol{e}}_0$
**for** $k$ *in* $0, \ldots, K$ **do**

$$\hat{\boldsymbol{B}}_{k|-k} := \left(\boldsymbol{H}\,\hat{\boldsymbol{P}}_{k|-k}\,\boldsymbol{H}^\top + \boldsymbol{R}\right)^{-1}$$

$$\hat{\boldsymbol{y}}_{k|-k} := \boldsymbol{H}\,\hat{\boldsymbol{x}}_{k|-k}$$

$$\mathcal{L}_k^{(i)} := \mathcal{L}_{k-1}^{(i)} + (\boldsymbol{y}_k - \hat{\boldsymbol{y}}_{k|-k})^\top \hat{\boldsymbol{B}}_{k|-k}\,(\boldsymbol{y}_k - \hat{\boldsymbol{y}}_{k|-k}) - \log \det \hat{\boldsymbol{B}}_{k|-k}$$

$$\hat{\boldsymbol{K}}_k := \hat{\boldsymbol{P}}_{k|-k}\,\boldsymbol{H}^\top \hat{\boldsymbol{B}}_{k|-k}$$

$$\hat{\boldsymbol{P}}_{k|-k} := \hat{\boldsymbol{P}}_{k|-k} - \hat{\boldsymbol{K}}_k\,\boldsymbol{H}\,\hat{\boldsymbol{P}}_{k|-k}$$

$$\hat{\boldsymbol{x}}_{k|-k} := \hat{\boldsymbol{x}}_{k|-k} + \hat{\boldsymbol{K}}_k\,(\boldsymbol{y}_k - \hat{\boldsymbol{y}}_{k|-k})$$

$$\boldsymbol{h}_{k+1}^{\rightarrow} := \mathrm{GRU}_\phi(\boldsymbol{h}_k^{\rightarrow}, \boldsymbol{y}_k$$

$$\boldsymbol{h}_{k+1}^{\leftarrow} := \mathrm{GRU}_\phi(\boldsymbol{h}_k^{\leftarrow}, \boldsymbol{y}_{k+1}$$

$$\begin{bmatrix} \hat{\boldsymbol{e}}_{k+1|-(k+1)} \\ \hat{\boldsymbol{F}}_{k+1|-(k+1)} \\ \hat{\boldsymbol{L}}_{k+1|-(k+1)} \end{bmatrix} := \boldsymbol{f}_\phi(\boldsymbol{h}_{k+1}^{\rightarrow}, \boldsymbol{h}_{k+1}^{\leftarrow})$$

$$\hat{\boldsymbol{Q}}_{k+1|-(k+1)} := \hat{\boldsymbol{L}}_{k+1|-(k+1)}\,\hat{\boldsymbol{L}}_{k+1|-(k+1)}^\top$$

$$\hat{\boldsymbol{P}}_{k+1|-(k+1)} := \hat{\boldsymbol{F}}_{k+1|-(k+1)}\,\hat{\boldsymbol{P}}_{k|0:K}\,\hat{\boldsymbol{F}}_{k+1|-(k+1)}^\top + \hat{\boldsymbol{Q}}_{k+1|-(k+1)}$$

$$\hat{\boldsymbol{x}}_{k+1|-(k+1)} := \hat{\boldsymbol{F}}_{k+1|-(k+1)}\,\hat{\boldsymbol{x}}_{k|0:K} + \hat{\boldsymbol{e}}_{k+1|-(k+1)}$$

**end**
For training case we also use backpropagation through the above loop to compute $\nabla_\phi \mathcal{L}_K$

---

**Algorithm 4:** Parameterized Recursive Smoother (Training)

---

**input** : Data (time-series) $\boldsymbol{y}_{0:K} = (\boldsymbol{y}_0, \ldots, \boldsymbol{y}_K)$, emission matrices $\boldsymbol{H}$ and $\boldsymbol{R}$, initialized parameters $\phi_0$, number of training rounds $I$

**output:** Model parameters $\phi^*$ for inference at test-time.

**for** $i$ *in* $1, \ldots, I$ **do**

Obtain $\nabla_\phi \mathcal{L}_K^{(i)}$ from algorithm 3.
Run preferred optimizer step.

**end**

---

---

**Algorithm 5:** Linearized Smoother

---

**input** : Values of: $\hat{\boldsymbol{F}}_{k|<k}, \hat{\boldsymbol{P}}_{k|<k}, \hat{\boldsymbol{P}}_{k|\leq k}, \hat{\boldsymbol{x}}_{k|\leq k}$ for all $k = 0, \ldots, K$, obtained from the recursive filter algorithm.

**output:** Linearly smoothed distributions $p(\boldsymbol{x}_k \mid \boldsymbol{y}_{0:K}) = \mathcal{N}\left(\boldsymbol{x}_k \mid \hat{\boldsymbol{z}}_k, \hat{\boldsymbol{G}}_k\right)$ for all $k = 0, \ldots, K$.

$\hat{\boldsymbol{z}}_K := \hat{\boldsymbol{x}}_{K|\leq K}$

$\hat{\boldsymbol{G}}_K := \hat{\boldsymbol{P}}_{K|\leq K}$

**for** $k = K, \ldots, 1$ **do**

$\quad \hat{\boldsymbol{J}}_{k-1|k} := \hat{\boldsymbol{P}}_{k-1|\leq k-1} \, \hat{\boldsymbol{F}}_{k|<k}^{\top} \, \hat{\boldsymbol{P}}_{k|<k}^{-1}$

$\quad \hat{\boldsymbol{G}}_{k-1} := \hat{\boldsymbol{P}}_{k-1|\leq k-1} + \hat{\boldsymbol{J}}_{k-1|k} \left(\hat{\boldsymbol{P}}_{k|\leq k} - \hat{\boldsymbol{P}}_{k|<k}\right) \hat{\boldsymbol{J}}_{k-1|k}^{\top}$

$\quad \hat{\boldsymbol{z}}_{k-1} := \hat{\boldsymbol{x}}_{k-1|\leq k-1} + \hat{\boldsymbol{J}}_{k-1|k} \left(\hat{\boldsymbol{z}}_k - \hat{\boldsymbol{x}}_{k|\leq k}\right)$

**end**

---

**Algorithm 6:** Recurrent Smoother (Training)

---

**input** : Training data (time-series) $\boldsymbol{y}_{0:K} = (\boldsymbol{y}_0, \ldots, \boldsymbol{y}_K)$, emission function $\mathcal{N}(\boldsymbol{H}\boldsymbol{x}_k, \boldsymbol{R})$, initialized parameters $\phi_0$, number of training iterations $n$.

**output:** Optimized parameters $\phi^*$

**for** *i in 1 to n* **do**

$\quad \boldsymbol{h}_0 := \boldsymbol{0}$

$\quad \boldsymbol{h}_{K+1} := \boldsymbol{0}$

$\quad \mathcal{L}^{(i)} := 0$

$\quad$ **for** *k in 0 to K* **do**

$\qquad \boldsymbol{h}_k^{\rightarrow} := \mathrm{GRU}_\phi(\boldsymbol{h}_{k-1}^{\rightarrow}, \boldsymbol{y}_{k-1})$

$\qquad \boldsymbol{h}_k^{\leftarrow} := \mathrm{GRU}_\phi(\boldsymbol{h}_{k-1}^{\leftarrow}, \boldsymbol{y}_{k+1})$

$\qquad \begin{bmatrix} \hat{\boldsymbol{x}}_{k|-k} \\ \hat{\boldsymbol{L}}_{k|-k} \end{bmatrix} := \boldsymbol{f}_\phi(\boldsymbol{h}_k^{\leftarrow}, \boldsymbol{h}_k^{\rightarrow})$

$\qquad \hat{\boldsymbol{P}}_{k|-k} := \hat{\boldsymbol{L}}_{k|-k} \hat{\boldsymbol{L}}_{k|-k}^{\top}$

$\qquad \hat{\boldsymbol{y}}_{k|-k} := \boldsymbol{H}\hat{\boldsymbol{x}}_{k|-k}$

$\qquad \hat{\boldsymbol{B}}_{k|-k} := \left(\boldsymbol{H}\hat{\boldsymbol{P}}_{k|-k}\boldsymbol{H}^{\top} + \boldsymbol{R}\right)^{-1}$

$\qquad \mathcal{L}_k^{(i)} := \mathcal{L}_{k-1}^{(i)} + (\boldsymbol{y}_k - \hat{\boldsymbol{y}}_{k|-k})^{\top} \hat{\boldsymbol{B}}_{k|-k} (\boldsymbol{y}_k - \hat{\boldsymbol{y}}_{k|-k}) - \log \det \hat{\boldsymbol{B}}_{k|-k}$

$\quad$ **end**

$\quad$ Compute $\nabla_\phi \mathcal{L}_K^{(i)}$ and apply SGD step with respect to all model parameters, which amounts to backpropagation through the above calculations.

**end**

*For the filter variant, the steps that involve the backward direction ($\leftarrow$) are left out.*

---

---

**Algorithm 7:** Recurrent Smoother (Inference)

---

**input** : Test data $\boldsymbol{y}_{0:K} = (\boldsymbol{y}_0, \ldots, \boldsymbol{y}_K)$, trained parameters $\phi^*$, emission function $\mathcal{N}(\boldsymbol{H}\boldsymbol{x}, \boldsymbol{R})$.

**output:** Inferred posteriors $p(\boldsymbol{x}_k \mid \boldsymbol{y}_{0:K}) = \mathcal{N}(\boldsymbol{x}_k \mid \hat{\boldsymbol{x}}_{k|0:K}, \hat{\boldsymbol{P}}_{k|0:K})$ for all $k$

$\boldsymbol{h}_0 := \boldsymbol{0}$

$\boldsymbol{h}_{K+1} := \boldsymbol{0}$

**for** *k in 0 to K* **do**

$$\boldsymbol{h}_k^{\rightarrow} := \mathrm{GRU}_\phi(\boldsymbol{h}_{k-1}^{\rightarrow}, \boldsymbol{y}_{k-1})$$

$$\boldsymbol{h}_k^{\leftarrow} := \mathrm{GRU}_\phi(\boldsymbol{h}_{k-1}^{\leftarrow}, \boldsymbol{y}_{k+1})$$

$$\begin{bmatrix} \hat{\boldsymbol{x}}_{k|-k} \\ \hat{\boldsymbol{L}}_{k|-k} \end{bmatrix} := \boldsymbol{f}_\phi(\boldsymbol{h}_k^{\leftarrow}, \boldsymbol{h}_k^{\rightarrow})$$

$$\hat{\boldsymbol{P}}_{k|-k} := \hat{\boldsymbol{L}}_{k|-k}\hat{\boldsymbol{L}}_{k|-k}^{\top}$$

$$\hat{\boldsymbol{K}}_k := \hat{\boldsymbol{P}}_{k|-k}\boldsymbol{H}^{\top}\left(\boldsymbol{H}\hat{\boldsymbol{P}}_{k|-k}\boldsymbol{H}^{\top} + \boldsymbol{R}\right)^{-1}$$

$$\hat{\boldsymbol{x}}_{k|0:K} := \hat{\boldsymbol{x}}_{k|-k} + \hat{\boldsymbol{K}}_k(\boldsymbol{y}_k - \boldsymbol{H}\hat{\boldsymbol{x}}_{k|-k})$$

$$\hat{\boldsymbol{P}}_{k|0:K} := \hat{\boldsymbol{P}}_{k|-k} - \hat{\boldsymbol{K}}_k\boldsymbol{H}\hat{\boldsymbol{P}}_{k|-k}$$

**end**

*For the filter variant, the steps that involve the backward direction ($\leftarrow$) are left out.*

---

## F  EXPERIMENTS: DETAILS

### F.1  LINEAR DYNAMICS

As specified in the main paper, the dynamics are according to

$$\dot{\boldsymbol{x}} = \boldsymbol{A}\boldsymbol{x} = \begin{bmatrix} 0 & 1 & 0 \\ 0 & -c & 1 \\ 0 & -\tau c & 0 \end{bmatrix} \begin{bmatrix} p \\ v \\ a \end{bmatrix}. \tag{66}$$

Since these are linear transitions, we can calculate any transition directly using $\boldsymbol{x}(t + \Delta t) = e^{\boldsymbol{A}\Delta t}\boldsymbol{x}(t)$.

$$\boldsymbol{F} := \begin{bmatrix} e^{\boldsymbol{A}} & 0 \\ 0 & e^{\boldsymbol{A}} \end{bmatrix} \qquad\qquad \boldsymbol{Q} := \begin{bmatrix} \bar{\boldsymbol{Q}} & 0 \\ 0 & \bar{\boldsymbol{Q}} \end{bmatrix} \tag{67}$$

We used $c = 0.06, \tau = 0.17, \Delta t := 1$ and covariance

$$\bar{\boldsymbol{Q}} := 0.1^2 \cdot \begin{bmatrix} \frac{1}{3} & 0 & 0 \\ 0 & 1 & 0 \\ 0 & 0 & 3 \end{bmatrix} \tag{68}$$

The matrix exponential is computed using Bader et al. (2019). The parameters for the emission distribution:

$$\boldsymbol{H} := \begin{bmatrix} 1 & 0 & 0 & 0 & 0 & 0 \\ 0 & 0 & 0 & 1 & 0 & 0 \end{bmatrix} \qquad\qquad \boldsymbol{R} := 0.5^2 \cdot \begin{bmatrix} 1 & 0 \\ 0 & 1 \end{bmatrix} \tag{69}$$

We simulate a $K := 131,072$ trajectory for training, $K := 16,384$ trajectory for validation and $K := 32,768$ for testing. The $\tilde{\boldsymbol{F}}$ that is used in the (non-optimal) Kalman filter and recursive model is computed as follows:

$$e^{\boldsymbol{A}\Delta t} \approx \tilde{\boldsymbol{F}} := \sum_{n=0}^{1} (\Delta t \boldsymbol{A}^n)/n! \tag{70}$$

### F.2  LORENZ EQUATIONS

We simulate a Lorenz system according to

$$\dot{\boldsymbol{x}} = \boldsymbol{A}\boldsymbol{x} = \begin{bmatrix} -\sigma & \sigma & 0 \\ \rho - x_1 & -1 & 0 \\ x_2 & 0 & -\beta \end{bmatrix} \begin{bmatrix} x_1 \\ x_2 \\ x_3 \end{bmatrix}. \tag{71}$$

We integrate the system using $dt = 0.00001$ and sample it uniformly at $\Delta t = 0.05$. We use $\rho = 28$, $\sigma = 10$, $\beta = 8/3$. The transition in $\Delta t$ arbitrary time-steps is linearly approximated by a Taylor expansion and used in the Kalman smoother and recursive models.

$$e^{\boldsymbol{A}_{|\boldsymbol{x}_k}\Delta t} \approx \tilde{\boldsymbol{F}}_k := \sum_{n=0}^{2} (\Delta t \boldsymbol{A}_{|\boldsymbol{x}_k})^n/n! \tag{72}$$

We simulate $K := 131,072$ steps for training, $K := 32,768$ for testing and $K := 16,384$ for validation. We have $\boldsymbol{H} := \boldsymbol{I}$ and thus $\boldsymbol{x} \in \mathbb{R}^3$ and $\boldsymbol{y} \in \mathbb{R}^3$. We use $\boldsymbol{R} := 0.5^2 \boldsymbol{I}$.

## G  PARAMETERIZED SMOOTHING: ALTERNATIVE POSTERIOR EVALUATION

We would like to point out that the distribution $p(\boldsymbol{x}_k \mid \boldsymbol{y}_{0:K})$ can be obtained without making assumption eq. (30), which we stretch out here. Initial experiments showed that using these calculations the model did not converge as smoothly as when using the ones stated before. However, it could be of interest to further investigate. Returning to the posterior of interest

$$p(\boldsymbol{x}_k \mid \boldsymbol{y}_{0:K}) = \int p(\boldsymbol{x}_k \mid \boldsymbol{x}_{k-1}, \boldsymbol{y}_{0:K})\, p(\boldsymbol{x}_{k-1} \mid \boldsymbol{y}_{0:K})\, d\boldsymbol{x}_{k-1} \tag{73}$$

$$= \int \frac{p(\boldsymbol{y}_k \mid \boldsymbol{x}_k)}{p(\boldsymbol{y}_k \mid \boldsymbol{x}_{k-1}, \boldsymbol{y}_{-k})}\, p(\boldsymbol{x}_k \mid \boldsymbol{x}_{k-1}, \boldsymbol{y}_{-k}) p(\boldsymbol{x}_{k-1} \mid \boldsymbol{y}_{0:K})\, d\boldsymbol{x}_{k-1}. \tag{74}$$

We have

$$p(\boldsymbol{x}_{k-1} \mid \boldsymbol{y}_{0:K}) = \mathcal{N}\left(\boldsymbol{x}_{k-1} \mid \hat{\boldsymbol{x}}_{k-1|0:K}(\boldsymbol{y}_{0:K}), \hat{\boldsymbol{P}}_{k-1|0:K}(\boldsymbol{y}_{0:K})\right) \tag{75}$$

as the previous time-step's posterior.

$$p\left(\boldsymbol{x}_k \mid \boldsymbol{x}_{k-1}, \boldsymbol{y}_{-k}\right) = \mathcal{N}\left(\boldsymbol{x}_k \mid \hat{\boldsymbol{F}}_{k|-k}\boldsymbol{x}_{k-1} + \hat{\boldsymbol{e}}_{k|-k}, \hat{\boldsymbol{Q}}_{k|-k}\right) \tag{76}$$

where $\hat{\boldsymbol{F}}_{k|-k}(\boldsymbol{y}_{-k})$, $\hat{\boldsymbol{e}}_{k|-k}(\boldsymbol{y}_{-k})$ and $\hat{\boldsymbol{Q}}_{k|-k}(\boldsymbol{y}_{-k})$ are estimated by a neural network. Combining this with noise model eq. (3) we get:

$$p(\boldsymbol{x}_k \mid \boldsymbol{x}_{k-1}, \boldsymbol{y}_{0:K})$$
$$= \mathcal{N}\left(\boldsymbol{x}_k \mid \hat{\boldsymbol{F}}_{k|-k}\,\boldsymbol{x}_{k-1} + \hat{\boldsymbol{e}}_{k|-k} + \boldsymbol{K}_k\left(\boldsymbol{y}_k - \boldsymbol{H}\left(\hat{\boldsymbol{F}}_{k|-k}\,\boldsymbol{x}_{k-1} + \hat{\boldsymbol{e}}_{k|-k}\right)\right), \hat{\boldsymbol{Q}}_{k|-k} - \boldsymbol{K}_k\,\boldsymbol{H}\,\hat{\boldsymbol{Q}}_{k|-k}\right) \tag{77}$$

$$= \mathcal{N}\left(\boldsymbol{x}_k \mid (\boldsymbol{I} - \boldsymbol{K}_n\,\boldsymbol{H})\left(\hat{\boldsymbol{F}}_{k|-k}\,\boldsymbol{x}_{k-1} + \hat{\boldsymbol{e}}_{k|-k}\right) + \boldsymbol{K}_k\,\boldsymbol{y}_k, (\boldsymbol{I} - \boldsymbol{K}_n\,\boldsymbol{H})\,\hat{\boldsymbol{Q}}_{k|-k}\right), \tag{78}$$

where we directly applied the Woodbury matrix identity to obtain Kalman gain matrix

$$\boldsymbol{K}_k := \hat{\boldsymbol{Q}}_{k|-k}\,\boldsymbol{H}^{\top}\left(\boldsymbol{H}\,\hat{\boldsymbol{Q}}_{k|-k}\,\boldsymbol{H}^{\top} + \boldsymbol{R}\right)^{-1} \tag{79}$$

Then, applying the integral we get:

$$p\left(\boldsymbol{x}_k \mid \boldsymbol{y}_{0:K}\right) = \mathcal{N}\left(\boldsymbol{x}_k \mid \hat{\boldsymbol{x}}_{k|0:K}, \hat{\boldsymbol{P}}_{k|0:K}\right), \tag{80}$$

with:

$$\hat{\boldsymbol{P}}_{k|0:K} = (\boldsymbol{I} - \boldsymbol{K}_n\,\boldsymbol{H})\,\hat{\boldsymbol{F}}_{k|-k}\,\hat{\boldsymbol{P}}_{k-1|0:K}\,\hat{\boldsymbol{F}}_{k|-k}^{\top}\,(\boldsymbol{I} - \boldsymbol{K}_n\,\boldsymbol{H})^{\top} + (\boldsymbol{I} - \boldsymbol{K}_n\,\boldsymbol{H})\,\hat{\boldsymbol{Q}}_{k|-k}, \tag{81}$$

and

$$\hat{\boldsymbol{x}}_{k|0:K} = \hat{\boldsymbol{F}}_{k|-k}\hat{\boldsymbol{x}}_{k-1|0:K} + \hat{\boldsymbol{e}}_{k|-k} + \boldsymbol{K}_k\left(\boldsymbol{y}_k - \boldsymbol{H}\left(\hat{\boldsymbol{F}}_{k|-k}\hat{\boldsymbol{x}}_{k-1|0:K} + \hat{\boldsymbol{e}}_{k|-k}\right)\right) \tag{82}$$

$$= (\boldsymbol{I} - \boldsymbol{K}_n\,\boldsymbol{H})\left(\hat{\boldsymbol{F}}_{k|-k}\,\hat{\boldsymbol{x}}_{k-1|0:K} + \hat{\boldsymbol{e}}_{k|-k}\right) + \boldsymbol{K}_k\,\boldsymbol{y}_k. \tag{83}$$

