# OpenReview forum: "Self-Supervised Inference in State-Space Models"
_ICLR.cc/2022/Conference — ICLR 2022 Poster_

### Official Review · Reviewer_q53Q · 2021-10-23

**Correctness:** 4
**Technical Novelty And Significance:** 3
**Empirical Novelty And Significance:** 2
**Recommendation:** 6
**Confidence:** 4

**Main Review:**

The manuscript's proposal is reasonable and interesting, but I have concerns about novelty. Also, I found some of the discussion a bit confusing, and the overall organization of the paper can be further improved.

Below are more specific comments :

1. Introduction : The opening paragraph suddenly loads a lot of references to the reader, without really putting them into a context. I would suggest selecting some of them and going into a bit of a more detail, if they are helpful in describing the approach put forth in the manuscript.

2. Introduction : I was also expecting a more concrete description of the proposal in the introduction. You are proposing to replace some of the parameters/matrices that occur in the Kalman filter with neural networks, which is perfectly fine, and is a simple idea (simple=good!). It would help to show the Kalman filter formulation (i.e., what it's solving), and outline what you're doing differently, without having to go into heavy notation.

3. Introduction : in the list of the contributions, for the first item : can you make that into a precise statement (like a proposition) and refer to it in the paper, so it's clear what you're showing? Are you referring to the development in Section 3? If so, I don't see what's novel with that development (other than replacing the said matrices with observation dependent estimates obtained through neural networks) -- can you elaborate?

4. Eqn 6 : What's the difference between the first line and the second line? Are you just replacing $<k$ with $\leq (k-1)$? If so, I'd suggest removing this equation. In fact, since $<$ and $\leq$ appear as subscripts, it may be good to stick to either one of them, so as to avoid confusion.

5. As I noted above, I think the proposal is simple, but I'd expect that training the neural networks might be a challenge in practice (if not, that's great). However, the idea reminded me of the paper titled "AI-IMU Dead Reckoning", by Brossard, Barrau, Bonnabel (https://arxiv.org/pdf/1904.06064.pdf) -- see for instance the comments right before section III.A. Can you point out the departure of your idea from what Brossard et al. do in their paper?

6. Another reference, which might be good to place the proposal into context is "Discriminative training of Kalman filters", by Abbeel, Coates, Montemerlo, Ng, Thrun, 2005. That work is certainly different from what you're doing, but also aims to find a good fixed set of covariance matrices from data.

7. Audio Denoising Experiment : Is it possible to train the network on a portion of the signal, and then stop training, so as to switch to "test" mode? More generally, I found the description of this experiment a bit terse. I'd welcome it if you can give a more detailed description, possibly using equations to mark what the input/output or the noise term etc. is.

**Summary Of The Paper:**

A Kalman filter typically requires specifying a state update equation, along with noise covariance matrices for state transition and observation. Determining these can be a challenge in certain scenarios. The manuscript proposes to replace these unknowns with trainable neural networks. The input to the neural networks are the observations, and the output are the unknown parameters (or their factorizations, to ensure positive definiteness). The manuscript validates the approach through numerical experiments.

**Summary Of The Review:**

Overall, the manuscript discusses an interesting approach, and has wide application potential, given that it promises to facilitate the tuning of Kalman filters. However, I have concerns about novelty. I also think the paper could be better organized.

---

> ### Author Response · Authors · 2021-11-23
> **Comment on Official Review of Paper2992 by Reviewer q53Q (1)**
>
>  **Summary**
> The reviewer has concerns about novelty and structure. Furthermore, some of the discussion was slightly confusing. Despite these, it is stated that the application potential is wide and that the approach is reasonable and interesting. Moreover, the reviewer considers the unsophisticated nature of the work a good thing.
>
> **Comments**
> We thank the reviewer for the constructive, actionable feedback. We have tried our best to accommodate the reviewer's points of concern regarding the latter (detailed below); we, unfortunately, cannot do much about the former other than provide arguments.
>
> It seems that the reviewer summarizes the approach as merely a way to parameterize a Kalman filter. While the final parameterizations are indeed almost identical, we think there is more to it for the following three reasons.
> 1. The "recurrent model" proposed in appendix C directly parameterizes $p(x_k \mid y_{<k})$ with neural networks. This model (similar to Lehtinen et al., 2018) allows for very flexible estimates of said probability. While the model proposed in the body of the paper seems restricted to just parameterizing a Kalman filter, it actually *reduces* to this model by putting $\hat{F}_k:=0$ for all $k$. Hence, it does not necessarily have the limitations of the hidden Markov process assumption of the Kalman filter. Since we do not make such assumptions, the model foundationally is different, even though the final parameterizations are highly similar.
> 2. The fact that we can estimate the posterior $p(x_k \mid y_{\leq k})$ analytically while making use of nonlinear neural networks was an interesting observation to us.
> 3. The Noise2Noise provably allows us to obtain clean estimates of $x_k$ using neural networks in the absence of clean data, which paired nicely with our neurally estimated transition probabilities.
>
> These points were probably not clear enough from the paper, understandably leading to the reviewer's concern. Hence, we have included similar arguments in the introduction.

---

> > ### Author Response · Authors · 2021-11-23
> > **Comment on Official Review of Paper2992 by Reviewer q53Q  (2)**
> >
> > Finally, we address the specific comments in the proposed order.
> > 1. *References are too sudden without contextualization.* We agree that there is a lack of context here. Reviewer d1x8 also points out that related work deserves its own section with more detail on earlier approaches. This we followed, leaving the introduction to "sketch the scene" better. Please consider the first paragraph of the revised manuscript.
> > 2. *Better description of the proposal in the introduction*. Since we now have a separate related-work section, there is room to do precisely what the reviewer suggests, i.e., be more explicit about the proposal and its relation to classical filtering. We also include some of the arguments above here. Please see the "Related Work" section in the manuscript.
> > 3. *Alternative phrasing of contribution 1; better description of the novelty*. We updated this in the manuscript (see contribution 1 in the introduction). Indeed, the mentioned development is not novel. However, the fact that we can estimate the posterior filtering distribution $p(x_k \mid y_{\leq k})$ analytically while making use of neural function estimators was an interesting observation. We expected that the reviewers would not blindly have taken our word for it if we did not provide the development.
> > 4. *Redundant equation*. Indeed, it was just the notation. We decided to include that line since it clarifies why the resulting distribution is given by equations 6 and 7. We removed the line in the updated manuscript (equation 6).
> > 5. *Comparison to Brossard et al., 2019*. We thank the reviewer for the referred material; we did not know about its existence while writing the paper. Some observations: (1) The section right before III-A describes the standard Extended Kalman filter. In the final sentence, the authors propose to learn noise parameters $Q_n$ and $N_n$. (2) The transition and emission functions $f$ and $g$ are given by the domain-specific algorithm in sections III-A and III-B. (3) In section IV-A, it turns out that only $N_n$ is learned dynamically. From these, we conclude that, indeed, learning covariance $N_n$ with neural networks is similar to what our methods propose. Though, it is not entirely clear what their objective function is. Contrastingly, our paper starts from a rather high-level interest in estimating filtering posterior $p(x_k \mid y_{\leq k})$. From there, we develop a procedure for obtaining it even though the data-generating process can be highly nonlinear. A maximum likelihood objective emerges. We think this presentation is more general and principled than the referred paper's. We also like to mention that, contrary to what the reviewer suspects, the proposed models were not hard to train, as the objective is maximum likelihood.
> > 6. *Compare the work to Abbeel et al., 2005*. We thank the reviewer for the referred paper. In it, several approaches to learning Kalman filter covariance parameters $R$ and $S$ are proposed. We contrast our approach against theirs. First, the authors assume a Markov chain in latent space; our approach does not. Secondly, the transition and emission parameters are assumed to be constant for the whole generative process. Our approach instead dynamically adapts these based on observed data. Finally, only the covariance matrices $R$ and $S$ are considered, whereas our approach also considers the functional parameters.
> > 7. *Is it possible to train on part of the signal? Make section 8.3 less terse.* Yes, this is certainly possible. We have attempted to clarify the section (see 8.3) by writing the data products mathematically and more clearly specified the model's task.
> >
> > Again, we thank the reviewer for the helpful feedback. We tried to be as transparent as possible about the adjustments made to the manuscript, so that the reviewer can verify easily. Considering these and the arguments above, we hope that the reviewer is open to adjusting the score. If any additional questions or comments may arise, we are happy to discuss further.

---

> > > ### Comment · Reviewer_q53Q · 2021-11-28
> > > **response to authors**
> > >
> > > Thank you for the detailed responses. You have addressed most of my comments. I'm still a bit concerned about novelty, but I'd rather see this paper published than not, so I raised my score.

---

### Official Review · Reviewer_7PDK · 2021-10-29

**Correctness:** 4
**Technical Novelty And Significance:** 3
**Empirical Novelty And Significance:** 3
**Recommendation:** 8
**Confidence:** 3

**Main Review:**

strengths:
- The authors proposed a new inference model, and demonstrated the effectiveness of their method through solid quantitative comparisons in the experiment section. The paper is clear written.

weakness:
- Is it possible to show plots on the estimated latent trajectory for the real audio denoising data? (e.g. plots like fig.1 but for real data) I'm curious about what are the latent states look like for different noise groups? Are there any qualitative differences among the latent trajectory estimated for different methods besides the quantitative differences on MSE?



**Summary Of The Paper:**

The authors proposed a new inference method for state-space model with nonlinear latent states. They showed that their inference model can be incorporated with domain knowledge and optimized in a self-supervised manner. They also demonstrated that their model can achieve competitive results through both simulated and real audio denoising tasks.

**Summary Of The Review:**

The authors clearly state the contributions of their paper compared to the previous literature. The proposed inference method is novel, and the math derivations look solid. And the quantitative comparisons in the experiment results look competitive. Overall, I recommend this paper to be accepted.

---

> ### Author Response · Authors · 2021-11-23
> **Comment on Review of Paper2992 by Reviewer 7PDK**
>
> **Review Summary**
> The reviewer states that, compared to relevant literature, the inference model is novel and principled. The quantitative experiments are solid and demonstrate the effectiveness of the model.
>
> **Comments**
> We thank the reviewer for the overall positive feedback. We have uploaded plots (noise classes in the plot titles, baseline SIN provided for comparison). We are curious to hear whether the reviewer would deem it valuable to include these in the paper. Furthermore, by observing these plots, we realized that the vast majority of the audio contains silence, making for an uninteresting experiment. We updated the results in Table 2 by considering only the parts where actual words are spoken.
>
> We also included recordings of the audio experiment (white noise class) as supplementary material to the submission.  In the reconstructions, it can be heard that the noise is suppressed.
>
> Qualitatively, we found no differences between the models, except that the noise was less suppressed.
>
> Thanks again for the valuable feedback.

---

> > ### Comment · Reviewer_7PDK · 2021-11-29
> > **response to authors**
> >
> > I thank the authors for their responses. I found the latent plots useful, and you can consider including some of them in the paper appendix.

---

### Official Review · Reviewer_d1x8 · 2021-10-30

**Correctness:** 4
**Technical Novelty And Significance:** 3
**Empirical Novelty And Significance:** 2
**Recommendation:** 6
**Confidence:** 3

**Main Review:**

I found the idea and its connection with the noise2noise approach interesting. However, I have concerns about the empirical evidence of the work and some comments on the structure of the paper:
- Given that this area is an active area of research with a relatively large body of related work, I believe the related work deserves a separate sub/section. The current version of the paper has a list of related work in a long paragraph at the beginning of the paper. I suggest you move this to a separate section and provide a better overview of the relevant work (see “KalmanNet: Neural Network Aided Kalman Filtering for Partially Known Dynamics” for an example).
-  Noise2noise method has been mentioned and used as a baseline, but no description has been provided for this related idea. Adding a few sentences or a subsection in the appendix can help the reader unfamiliar with noise2noise to connect the works better.
- The “KalmanNet” paper, mentioned above seems to be closely related to the paper. As mentioned by the authors of KalmanNet, it can also be trained in an unsupervised fashion. Can you add the work to the "related works" section and describe the possible advantages of your recursive approach over theirs?
- The experiments provide evidence for 1) model working in simple synthetic examples (passing sanity tests) and 2) model performing better than KF and EKF in both synthetic and real dataset. However, a more extensive comparison with other unsupervised methods is recommended. Given at least 9 other unsupervised methods in Table 1, adding at least one more unsupervised baseline to the audio denoting experiment can be helpful. If none of these methods is applicable to the experiment, justification can be provided instead.
- The performance of SIN (Krishnan et al.) in figure 5 is a bit surprising to me. I’d expect a more gradual decline in MSE instead of a sharp drop. Can you provide some explanation for this behavior?

**Summary Of The Paper:**

The paper proposes an approximate inference framework for state-space models with nonlinear latent dynamics. The approach uses the classical Bayesian update rules with neural network for estimating the parameters of local linear transitions.

**Summary Of The Review:**

I found the proposed approach interesting; however, I think the paper can be further improved in terms of format and empirical evidence.
################### POST-REBUTTAL COMMENTS ########################

I found the authors' response adequate; hence, I'm adjusting my score.

---

> ### Author Response · Authors · 2021-11-23
> **Comment on Official Review of Paper2992 by Reviewer d1x8**
>
> **Summary**
> The reviewer requests more experimental validation. Formatting can also be improved. Despite these, the reviewer states that the approach is interesting, and connections to Noise2Noise are valuable.
>
> **Comments**
> We thank the reviewer for the helpful comments.  Regarding the novelty of the approach: please consider the first three arguments that we provided to reviewer q53Q, where we attempt to address this issue (omitted here for conciseness). Regarding the structure of the paper: we have now included a Related Work section (see point 1). The reviewer also requests more empirical evidence. Hence, we uploaded audio recordings and plots of the latent trajectories to the supplementary material of the submission. We address comments in the proposed order.
>
> 1. *Related work section is required.* We agree that a related work section works better. Hence, we included that in the updated manuscript (section 2). Other than that, we think that it is relatively complete. We thank the reviewer for the suggested work. Note that it was released after the initial version of this manuscript. Hence, it was overlooked. For additional comments, see 3.
> 2. *Better explanation of Noise2Noise method.* Since we struggle to keep within the page limit, we have included the reference where Noise2Noise is mentioned (see section 8.3). Is this satisfactory to the reviewer?
> 3. *Comparison to KalmanNet.* Indeed, the concurrently developed KalmanNet is highly similar to the proposed work. The authors indeed suggest a similar unsupervised approach as we did in our work but did not explicitly develop that idea. An additional difference is that our work seems more principled about the assumed generative process, the targeted filtering posterior $p(x_k \mid y_{\leq k})$, and the objective that arises naturally. Moreover, our paper includes *parameterized smoothing* (appendix A), *linearized smoothing* (section 5), and the *recurrent model* (appendix C). We also include theoretical guarantees under the Noise2Noise objective. Altogether, we deem these to be steps that go significantly further than the referred work. We have included the paper in our related work section, along with how our paper differs from theirs (see section 2, final sentences).
> 4. *Additional baseline for the audio experiment is requested.* Considering the unsupervised methods presented in Table 1, we see that Watter et al. (2015) does not perform explicit state estimation, and Doerr et al. (2015) use Gaussian processes, making their approach hard to scale to audio data. The remaining methods are all based on variational inference. During the rebuttal phase, we implemented SIN and ran the experiments. Results are presented in the table now—our model outperforms SIN on all noise classes. Moreover, SIN took significantly longer to train (\~3 days) than our models (\~1 day). Furthermore, we realized that the vast majority of the audio contains silence, making for an uninteresting experiment. We updated Table 2 by considering only the parts where actual words are spoken.
> 5. *Why does SIN perform poorly in Figure 5?*. Since VAEs both model a generative and an inference model, they require significantly more data than training just an inference model. Hence, they overfit heavily in the low-data regime. Due to its computational cost, we evaluated the SIN at fewer data regimes than the other models. Note that the $x$-axis is log-scale; the leap is from 256 data points to 4096, which is a sizable difference in available data. Finally, SIN assumes a Markov chain in state space in its generative network, which could be a limiting factor.  With enough data, SIN performs similarly to our models (see Figure 5).
>
> Again, we thank the reviewer for the helpful feedback. We tried to be as transparent as possible about the adjustments made to the manuscript, so that the reviewer can verify easily. Considering these and the arguments above, we hope that the reviewer is open to adjusting the score. If any additional questions or comments may arise, we are happy to discuss further.

---

### Official Review · Reviewer_SVPv · 2021-11-02

**Correctness:** 3
**Technical Novelty And Significance:** 3
**Empirical Novelty And Significance:** 2
**Recommendation:** 6
**Confidence:** 3

**Main Review:**

Major:
1) The authors keep calling the Kalman filter (and other variants such as the RKN) a 'supervised' method, but these methods do not need a 'ground truth' latent variables. They do, however, need a set of observations and a set of model parameters. It would be good for the authors to clarify what they mean and exactly what the differences are in their approach.
2) Needs more validation on real datasets, and more visualization for how it performs on the one real dataset that the authors did use. The RKN is seen to perform better than the proposed approach- is this also true for forecasting? What is the comparison between the estimated states x from different models?
3) The authors need to clarify the utility of their 'prior knowledge' - how much of the recursive filter results are due to this prior knowledge as opposed to the Bayesian filtering? Please separate out the contribution of the prior knowledge from the recursive filter.
4) The figures could be made much better. Figure 2 could apply to any state space model; the novel ideas take a while to get to.

Minor:
1) Please detail the difference between 'recurrent' and 'recursive' earlier, when you first introduce the terms in Figure 1. I would also recommend changing the terms to terms that are not as similar.
2) Please bold the values that perform the best in Table 2.

**Summary Of The Paper:**

The authors describe a parametrized inference approach for nonlinear dynamical models with linear observations. They borrow ideas from the well-known Kalman filter, and use Bayesian filtering and smoothing approaches to recursively estimate the states of the nonlinear model. The parameterization makes use of a locally linear transition between subsequent states using neural networks; thus the authors are still able to perform the Kalman-like filtering and smoothing.

**Summary Of The Review:**

The ideas in the manuscript are novel, but clarification is needed as stated in the Main Review. The figures need work. Moreover, additional empirical novelty, either further evaluation on the datasets already examined as a part of this manuscript, or on other datasets, is recommended.

---

> ### Author Response · Authors · 2021-11-23
> **Comment on Review of Paper2992 by Reviewer SVPv**
>
> **Summary**
> The reviewer states that some work has to be done on visualization. Also, there were some unclarities about the paper's claims. Despite these, the reviewer states that the ideas in the paper are novel and deems the paper above the acceptance threshold.
>
> **Comments**
> We thank the reviewer for the helpful feedback and positive recommendation. In the following comments, we include new visualizations and address unclarities.
> Major:
> 1. *Kalman filter and RKN are referred to as "supervised" methods.* The Kalman filter requires careful tuning its emission and transition functions along with the covariances. We found that optimizing these using the noisy data (using EM or maximum likelihood) resulted in even poorer performance than reported in the paper. Thus, we optimized the parameters using clean data and therefore the optimization was supervised. Note that this was mentioned in section 8.2 bullet point (2). We added clarification where required (for example, in section 8.3).
> 2. *Additional visualization and experimentation required.* We have uploaded plots (noise classes in the plot titles) and recordings of the audio experiment as supplementary material to the submission. We are curious to hear whether the reviewer would deem it valuable to include these in the paper. Qualitatively, we found no differences between the models, except that the noise was simply less suppressed. Regarding experimentation, we realized that the vast majority of the audio contains silence, making for an uninteresting experiment. We updated Table 2 by considering only the parts where actual words are spoken. We also added an additional baseline to the experiment (SIN). Its performance is also visualized in the plots in the supplementary material. Our model outperforms this model on all noise classes. Moreover, SIN took significantly longer to train (\~3 days) than our models (\~1 day). We argue that its relatively poor performance is due to the Markov assumption of the latent space, which is unrealistic for spoken audio. To see this, note that even a simple sine wave cannot be modeled effectively as a Markov chain. Hence, it regularizes the model too strongly for effective learning.
> 5. *Quantify the contribution of prior knowledge.* This can be seen in figures 4 and 5. The Recurrent Model does not incorporate expert knowledge and thus heavily overfits. The Recursive model, due to its inductive bias, can achieve a more optimal solution. The gap between these is the value that the expert knowledge provides.
> 6. *Value of Figure 2 is unclear*. For Section 3, Figure 2 is exemplary: it shows a generative model that has a higher-order Markov chain. Classical methods and some recent VI-based methods assume a generative model with a Markov chain in state-space. This means that these methods are sub-optimal for this kind of data. Our method, contrastingly, does not make this assumption. Furthermore, we re-use this figure to elaborate on choices made in Section 6. Note that Figures 4 and 5 are based on Satorras et al. (2019).
>
> Minor:
> 1. *Model names are confusing*. We had trouble finding appropriate model names, as many Kalman filter-related work has already coined such names. Would it help to add an identifier like Recursive Model (M1) and Recurrent Model (M2)? Other suggestions are also welcome. We have included the distinction in Figure 1 by adding references to the corresponding sections.
> 2. *Boldface the best-performing models*. We were hesitant to do so, as the supervised RKN model outperforms ours in most cases, which can confuse a reader. Would the reviewer still recommend doing so?
>
> Again, we thank the reviewer for the helpful feedback. We tried to be as transparent as possible about the adjustments made to the manuscript, so that the reviewer can verify easily. Considering these and the arguments above, we hope that the reviewer remains leaning towards acceptance. If any additional questions or comments may arise, we are happy to discuss further.

---

> > ### Comment · Reviewer_SVPv · 2021-12-01
> > **Response to authors**
> >
> > Thank you to the authors for the detailed response. I found the latents and the noise experiments somewhat helpful, so I would recommend including them in the supplementary material. The audio files are less useful.
> >
> > I will keep my initial score. Thanks again for responding thoroughly.

---

### Decision · Program_Chairs · 2022-01-20

**Decision:**

Accept (Poster)

**Comment:**

This paper presents a method for inference in state-space models with non-linear dynamics and linear-Gaussian observations. Instead of parameterizing a generative model, the paper proposes to parameterize the conditional distribution of current latent states given previous latent states and observations using locally linear transitions, where the parameters of the linear mappings are given by neural networks. Under fairly standard conditionally-independence assumptions, the paper uses known Bayesian filtering/smoothing tricks to derive a recursive estimation algorithm and a parameter-estimation method based on a simple maximum likelihood objective.

Overall, the reviewers found the idea to be novel and interesting and I agree.  They also found the relation to the noise2noise objective worth highlighting. Several concerns were raised during the discussion period, which I believe the authors addressed satisfactorily. However, I think the authors should bring the assumed distinction between ‘supervised’, ‘self-supervised’ and ‘unsupervised’ upfront, as usually these types of models are trained using the noisy data (to which the authors refer to as unsupervised).

Given the large body of literature on dynamical systems, filters and smoothers, I believe the paper will benefit significantly from more comparisons across a wider range of (and more realistic) datasets.